**Technical Report**

# Single-cell sortChIC identifies hierarchical chromatin dynamics during hematopoiesis

Peter Zeller[1,2,4], Jake Yeung [1,2,3,4], Helena Viñas Gaza [1,2], Buys Anton de Barbanson[1,2], Vivek Bhardwaj [1,2], Maria Florescu[1,2], Reinier van der Linden [1,2] & Alexander van Oudenaarden [1,2]✉

Post-translational histone modifications modulate chromatin activity to affect gene expression. How chromatin states underlie lineage choice in single cells is relatively unexplored. We develop sort-assisted single-cell chromatin immunocleavage (sortChIC) and map active (H3K4me1 and H3K4me3) and repressive (H3K27me3 and H3K9me3) histone modifications in the mouse bone marrow. During differentiation, hematopoietic stem and progenitor cells (HSPCs) acquire active chromatin states mediated by cell-type-specifying transcription factors, which are unique for each lineage. By contrast, most alterations in repressive marks during differentiation occur independent of the final cell type. Chromatin trajectory analysis shows that lineage choice at the chromatin level occurs at the progenitor stage. Joint profiling of H3K4me1 and H3K9me3 demonstrates that cell types within the myeloid lineage have distinct active chromatin but share similar myeloid-specific heterochromatin states. This implies a hierarchical regulation of chromatin during hematopoiesis: heterochromatin dynamics distinguish differentiation trajectories and lineages, while euchromatin dynamics reflect cell types within lineages.

Hematopoietic stem cells (HSCs) reside in the bone marrow (BM) and replenish diverse blood cell types[1,2]. During differentiation, hematopoietic stem and progenitor cells (HSPCs) restrict their potential to fewer lineages to yield mature blood cells[3]. These cell fate decisions have recently been dissected through single-cell mRNA sequencing (scRNA-seq) technologies[4–6].

The regulation of gene expression partially relies on post-translational modifications of histones that modulate chromatin activity[7,8]. Chromatin dynamics during hematopoiesis have been analyzed for accessible regions in single cells[9,10] and active chromatin marks in sorted blood cell types[11]. Although the role of repressive chromatin has been characterized in embryonic stem cells[12–15] and early development[16–18], repressive chromatin states during hematopoiesis have been unexplored.

The following two repressive chromatin states have a major role in gene regulation: a polycomb-repressed state, marked by H3K27me3 at gene-rich regions[19,20], and a heterochromatin state mainly found in gene-poor regions, marked by H3K9me3[16]. Conventional techniques to detect histone modifications involve chromatin immunoprecipitation (ChIP), which relies on affinity-purification of histone–DNA complexes. As immunoprecipitations are not feasible for single cells individually, protocols were developed that fragment and barcode single cells before pooling them for immunoprecipitation[21–23]. Alternatives to ChIP[24] circumvent this affinity-purification by using antibody tethering of either protein A-micrococcal nuclease (pA-MN)[24–28] or protein A-Tn5 transposase[29–34] that produce recoverable fragments only at the site of interest. Although these strategies allow profiling of histone modifications in single cells[31,32,34], they do not enrich for specific cell types,

¹Hubrecht Institute-KNAW (Royal Netherlands Academy of Sciences), Oncode Institute, Utrecht, The Netherlands. ²University Medical Center Utrecht, Utrecht, The Netherlands. ³Institute of Science and Technology Austria (ISTA), Klosterneuburg, Austria. ⁴These authors contributed equally: Peter Zeller, Jake Yeung. ✉e-mail: a.vanoudenaarden@hubrecht.eu

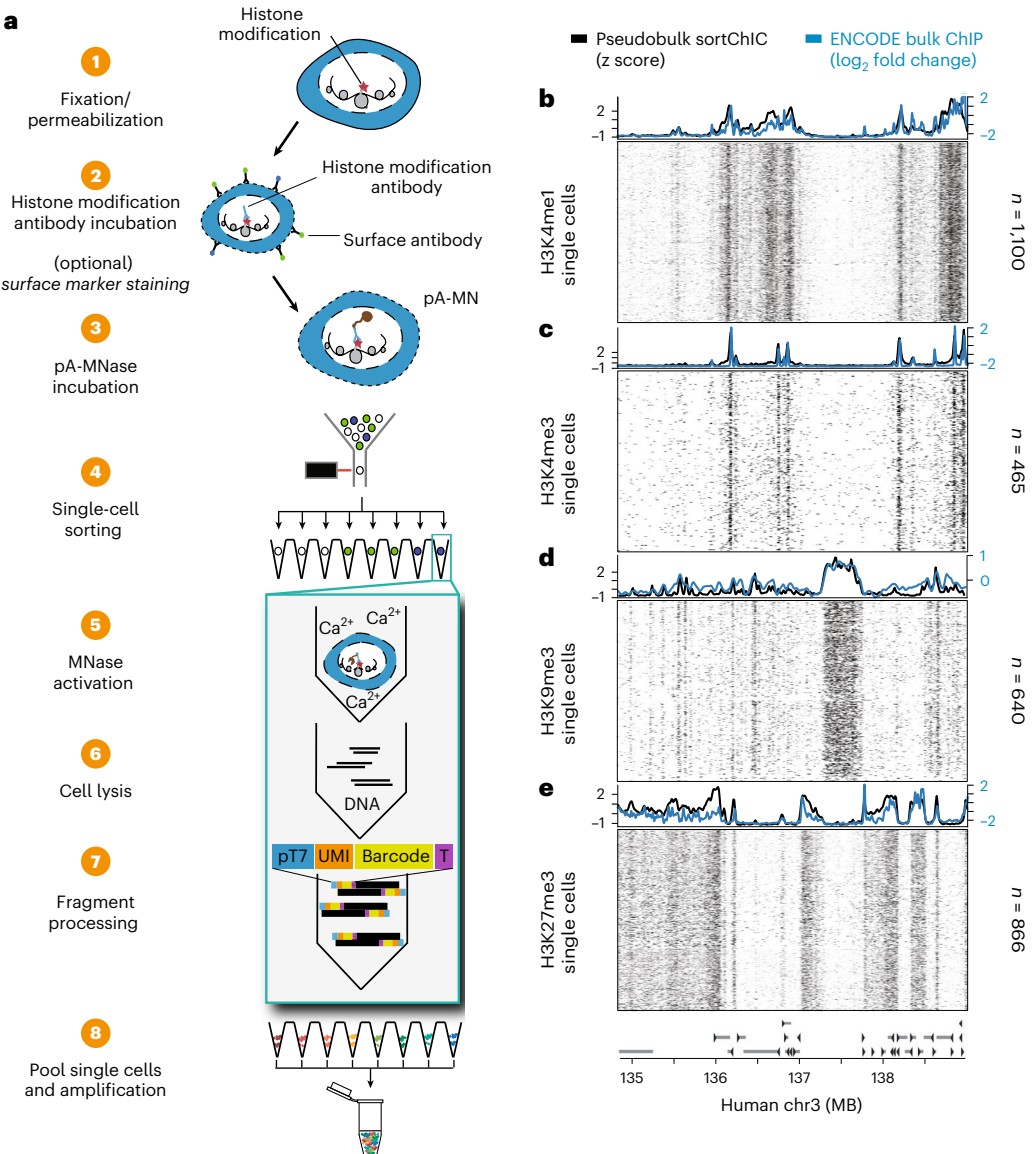

**Fig. 1 | sortChIC maps histone modifications in single cell. a**, Schematic of the sortChIC method. Fixed and permeabilized cells are stained with an antibody targeting a histone modification. Inactive pA-MN is added, tethering MN to the histone modification antibody. Single cells are FACS sorted. MN is activated to induce specific cuts in the genome. UMIs and cell-specific barcodes are ligated to the cut fragments. Barcoded fragments are pooled, amplified and sequenced. **b–e**, Location of cuts in H3K4me1 (**b**), H3K4me3 (**c**), H3K9me3 (**d**) and H3K27me3 (**e**) in individual K562 cells along a 4 MB region of chromosome three. Black traces represent the sortChIC signal averaged over all individual cells, blue traces represent ENCODE ChIP-seq profiles.

making it challenging to profile rare cell types, such as HSCs, that contribute about 0.01% of the cells[35]. Therefore, we develop sort-assisted single-cell chromatin immunocleavage (sortChIC), which combines single-cell histone modification profiling with cell enrichment.

## Results

### SortChIC maps histone modifications in single cells

To detect histone modifications in single cells, we stain surface antigens for cell type recognition, fix cells in ethanol and incubate them with an antibody against a histone modification. We then add pA-MN that binds to the histone-bound antibody at specific regions of the genome where the modification is present (Fig. 1a). Subsequently, single cells in G1 phase of the cell cycle are sorted based on their Hoechst staining into 384 well plates (Extended Data Fig. 1a). Next, MN is activated by adding calcium, allowing MN to digest antibody-proximal internucleosomal DNA regions. Removing the need for purification steps, nucleosomes

are digested and genomic DNA fragments are ligated to adapters containing a unique molecular identifier (UMI) and cell-specific barcode. The genomic fragments are amplified by in vitro transcription and PCR and sequenced.

To test sortChIC performance, we apply it to the well-characterized cell line K562, where we map four histone modifications that represent major chromatin states regulating gene expression (Fig. 1b–e). For modifications associated with gene activation, we profile H3K4me1 (Fig. 1b) and H3K4me3 (Fig. 1c), found at active enhancers and promoters and promoters of active genes, respectively[36]. For modifications associated with repression, we profile H3K9me3 found in gene-poor regions (Fig. 1d) and H3K27me3 found in gene-rich regions (Fig. 1e)[20].

For each histone modification, we process 1,128 G1 phase K562 cells. Using the MN cut site position and UMIs, we map unique MN cut sites. Following filtering, we retain 3,113 cells (Extended Data Fig. 1b) with the large majority of reads falling in peaks identified

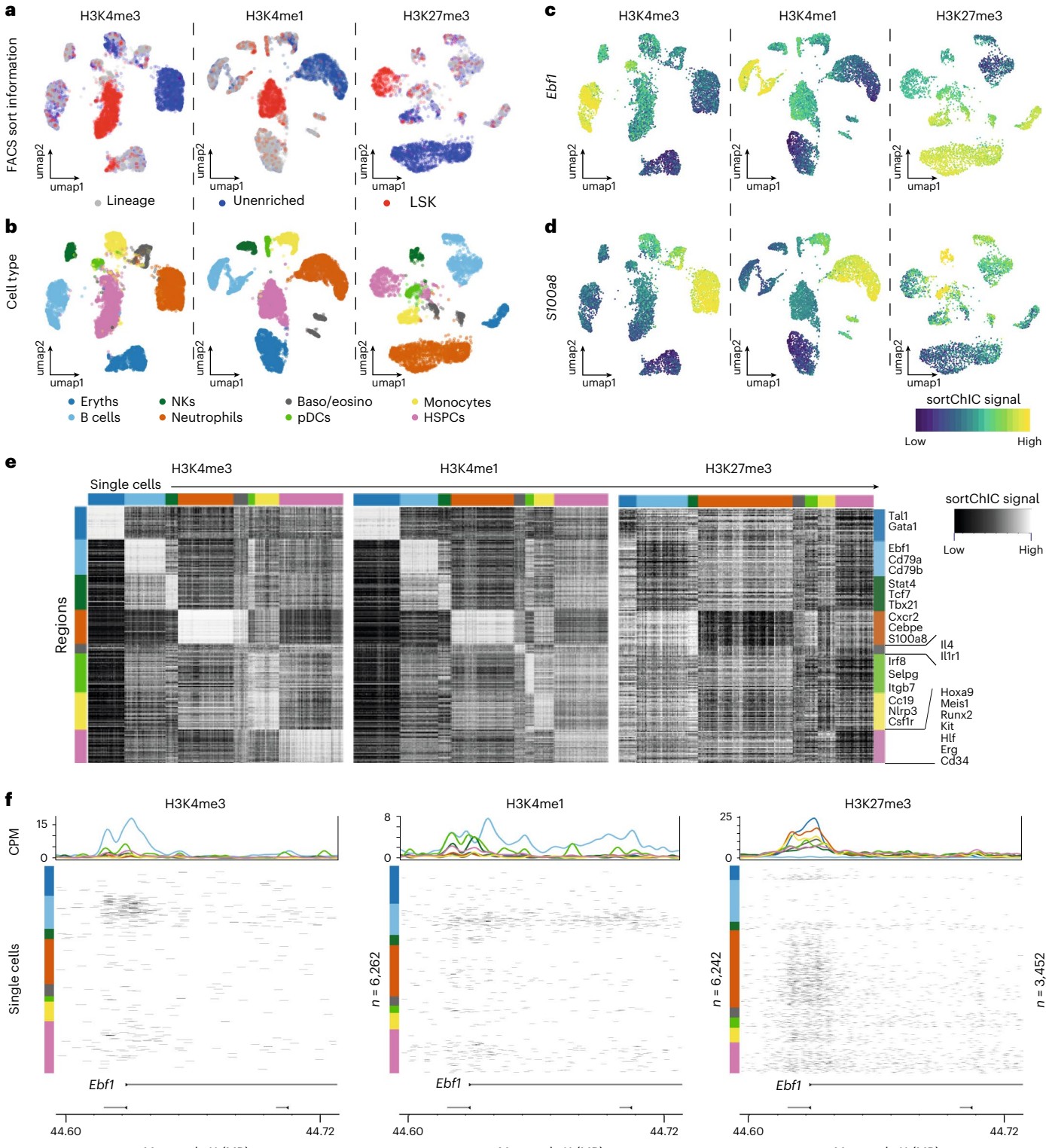

**Fig. 2 | Active and repressive chromatin states in single cells from the mouse BM. a**, UMAPs of H3K4me3 (*n* = 6,262), H3K4me1 (*n* = 6,242) and H3K27me3 (*n* = 3,452) single-cell epigenomes from whole BM (unenriched), Lin⁻ and LSK sorted populations. **b**, UMAPs colored by cell type. Eryths, erythroblasts; baso/ eosino, basophils/eosinophils; pDCs; monocytes; HSPCs, hematopoietic stem cells and early progenitor cells. **c**, UMAP summary colored by sortChIC signal in a region ±5 kb centered at the transcription start site of *Ebf1*, a B-cell-specific gene. **d**, Same as **c** but for a region around *S100a8*, a neutrophil-specific gene. **e**, Heatmap of sortChIC signals for regions around cell-type-specific genes showing high levels of active marks (H3K4me1, H3K4me3) in their respective cell type, and correspondingly low levels in the repressive mark (H3K27me3). **f**, Example of active and repressive chromatin states near the transcription start site of a B-cell-specific TF *Ebf1*. H3K4me3 and H3K4me1 show large number of cuts specifically in B cells; H3K27me3 shows B-cell-specific depletion of cuts. Colored line plots (same color code as in **b**) represent the average sortChIC signal for cells of the same cell type. Individual cells are ordered by cell type and color-coded on the left.

from pseudobulks (Extended Data Fig. 1c). We compare pseudobulk sortChIC profiles with bulk ChIP-seq results[37], which are highly correlated (Pearson correlation > 0.8; Extended Data Fig. 1d–e). Single-cell tracks underneath each average track (Fig. 1b–e) illustrate the high reproducibility of the signal between cells. Of note, the H3K9me3 histone modification profiles obtained from sortChIC represent the heterochromatin state without the need for input normalization (Extended Data Fig. 1f), which is required for ChIP experiments[38]. Lastly, we compare the sensitivity and specificity of sortChIC with existing methods. To compare sortChIC with pA-MN[22,27,28] and Tn5-based methods[30–32] (Extended Data Fig. 2a–c), we quantify sensitivity and signal specificity (Gini coefficient and signal enrichment). In terms of sensitivity, we find sortChIC to perform better than scChIP-seq and Tn5-based methods. While single-cell chromatin immunocleavage sequencing (scChIC-seq) and indexing single-cell immunocleavage sequencing (iscChIC-seq) have comparable or slightly higher sensitivity (Extended Data Fig. 2b,c, top left panel), both achieve this high signal at the expense of specificity (Extended Data Fig. 2b,c, bottom panels). A caveat for these comparisons is the use of different cell lines, antibodies and primary tissue samples.

### Active marks prime HSPCs, H3K27me3 marks mature alternatives

Next, we map active and repressive chromatin changes during blood formation. To equally include rare and common cell types from the mouse BM, we use cell surface markers Sca1, cKit and a set of lineage markers (Lin) to sort whole BM, lineage marker negative (Lin⁻) and LSK (Lin⁻Sca1⁺ckit⁺) cells that contain HSCs and multipotent progenitors (MPPs) and profile the same set of histone modifications (Extended Data Fig. 3a). Applying Latent Dirichlet Allocation (LDA)[39] and visualizing the output with Uniform Manifold Approximation and Projection (UMAP) reveals distinct clusters that contain LSKs, unenriched cell types or mixtures of lineage negative and unenriched cell types (Fig. 2a and Extended Data Fig. 3b). We use the H3K4me3 signal in promoter regions (transcription start site (TSS) ±5 kb) to determine marker genes for eight blood cell types (Fig. 2b). These regions contain known cell-type-specific genes such as the B-cell-specific transcription factor (TF), *Ebf1* (Fig. 2c), and the neutrophil-specific gene, *S100a8* (Fig. 2d). Specific regions are marked in a cell-type-dependent manner for H3K4me1 and H3K4me3. Conversely, these regions are depleted for H3K27me3 (Fig. 2e). This is exemplified by the TSS of the B-cell-specific TF, *Ebf1* (Fig. 2f). Next, we analyze published scRNA-seq data to determine mRNA abundances[4] associated with our cell-type-specific promoter regions and confirm that these sets of genes are cell-type-specific (Extended Data Fig. 3c). Interestingly, we find that HSPCs already have H3K4me3 and H3K4me1 signal at the *Ebf1* promoter and gene body suggesting HSPCs may already have active marks at genes before their expression in different lineages.

We extend the *Ebf1* observation to all TSSs in our eight cell-type-specific gene sets defined using H3K4me3, by comparing fold changes between differentiated cell type relative to HSPCs (Extended Data Fig. 3d–f). We find both up- and down-regulation of active chromatin. for example, at B-cell-specific genes, active chromatin levels increase from HSPCs to B cells and plasmacytoid dendritic cells (pDCs) but decrease in basophils/eosinophils, neutrophils and erythroblasts (Extended Data Fig. 3d,e). This divergence occurs in all eight cell-type-specific gene sets, suggesting that cell-type-specific

regions in HSPCs already have an intermediate level of active chromatin marks, which are modulated depending on the final cell type.

Repressive H3K27me3 at B-cell-specific genes, by contrast, is upregulated in nonB cells compared to HSPCs, while only few of them lose H3K27me3 signal upon B-cell differentiation (Extended Data Fig. 3f). Across other cell types, we observe a similar trend where mature cells upregulate H3K27me3 at genes specific for alternative cell fates, likely silencing cell type inappropriate genes.

In sum, our analysis of hematopoietic cell-type specific genes shows that in HSPCs active chromatin premarks genes of different blood cell fates, while H3K27me3 repressive chromatin during hematopoiesis silences genes of alternative fates.

### Dynamic H3K9me3 regions reveal HSPCs and three lineages

To understand chromatin regulation in heterochromatic regions, we explore H3K9me3. H3K9me3 analysis reveals the following four clusters: one cluster containing mostly LSKs, one containing mostly unenriched cells and two clusters containing a mixture of unenriched and lineage-negative cells (Fig. 3a,b). Large megabase-scale domains marked by H3K9me3 are constant across cell types; however, smaller regions display cluster-specific signals (Fig. 3c). Analysis of 50 kb regions across the genome identified 6,085 cluster-specific H3K9me3 regions ($q < 10^{-9}$, deviance goodness of fit). These regions have a 62.8 kb median distance to the nearest TSS, while noncluster-specific H3K9me3 regions have a 138 kb median distance to a TSS (Extended Data Fig. 4a). This suggests that cluster-specific H3K9me3 regions may be associated with gene regulation.

We hypothesize that H3K4me1 may also show differential enrichment in these cluster-specific H3K9me3 regions. Therefore, we select 150 regions with the largest depletion of the H3K9me3 compared to HSPC, resulting in four sets of cluster-specific regions (Extended Data Fig. 4b). The H3K4me1 signal in each of these four sets of regions shows cell-type-specific enrichment (Extended Data Fig. 4c), which anticorrelates with H3K9me3 (Fig. 3d). We use this anti-correlation to annotate H3K9me3-defined cell clusters as erythroid, lymphoid and myeloid lineages (Fig. 3e). We find that regions depleted of H3K9me3 in HSPCs show upregulation of H3K4me1 in HSPCs (Fig. 3f). For H3K9me3-depleted regions in myeloid cells, we find that H3K4me1 is upregulated not only in neutrophils but also in other cell types that share the myeloid lineage, such as monocytes (Fig. 3g). This anti-correlation is exemplified at the *Gbe1* locus. In this region, HSPCs, lymphoid and myeloid cell types show enrichment of H3K4me1 accompanied by a marked depletion in H3K9me3 (Fig. 3h). At these H3K9me3 regions, we also detect cell-type-specific signal in H3K4me3 and in H3K27me3, although the pattern is weaker than in H3K4me1 (Extended Data Fig. 4d). Overall, we find fewer cell clusters with distinguishable H3K9me3 distribution compared to active chromatin marks. We show that this reduction is the consequence of cell types of the same lineage sharing the same H3K9me3 signal.

### Repressive chromatin changes are mostly cell fate-independent

We next ask whether global patterns in chromatin dynamics during hematopoiesis differ between repressive and active marks. We apply differential analysis on 50 kb regions for all four marks, resulting in 10,518 dynamic bins for H3K4me1, 2,225 for H3K4me3, 5,494 for H3K27me3 and 6,085 for H3K9me3 (Supplementary Table 1). For each histone modification, we count the cell type pseudobulk signal across

---

**Fig. 3 | Heterochromatin state dynamics during hematopoiesis. a**, UMAP of H3K9me3 (*n* = 3,631) representing single cells from whole BM (unenriched), Lin⁻ and LSK sorted cells. **b**, Fraction of unenriched, Lin⁻ and LSK cells in each of the four H3K9me3 clusters. **c**, Region showing the H3K9me3 pseudobulk sortChIC signal of the four clusters. **d**, Heatmap of 50 kb bins displaying the relative H3K9me3 (left) and H3K4me1 (right) sortChIC signal in erythroblasts, lymphoid, myeloid and HSPCs. **e**, UMAP of H3K9me3 and H3K4me1 sortChIC data, colored

by cell type. **f**, Single-cell signal of cluster1-depleted bins (averaged across the 150 bins) showing low H3K9me3 and high H3K4me1 signal in lymphoid cells. Same bin set was used for both histone modifications. **g**, Single-cell signal of cluster3-specific bins showing low H3K9me3 and high H3K4me1 signal in myeloid cells. **h**, Zoom-in of the same genomic region in **c** for H3K9me3 and H3K4me1 pseudobulk sortChIC signal.

the bins and perform hierarchical clustering. In active marks, we find that the largest differences come from erythroblast versus nonerythroblasts (Extended Data Fig. 5a). This observation is consistent with the TSS analysis, where the erythroblasts show the largest changes in

active chromatin (Extended Data Fig. 3d–e). In accordance with the same TSS-centric analysis, we find intermediate levels of H3K4me1 and H3K4me3 in HSPCs (Extended Data Fig. 5a), suggesting a more accessible chromatin state in HSPCs.

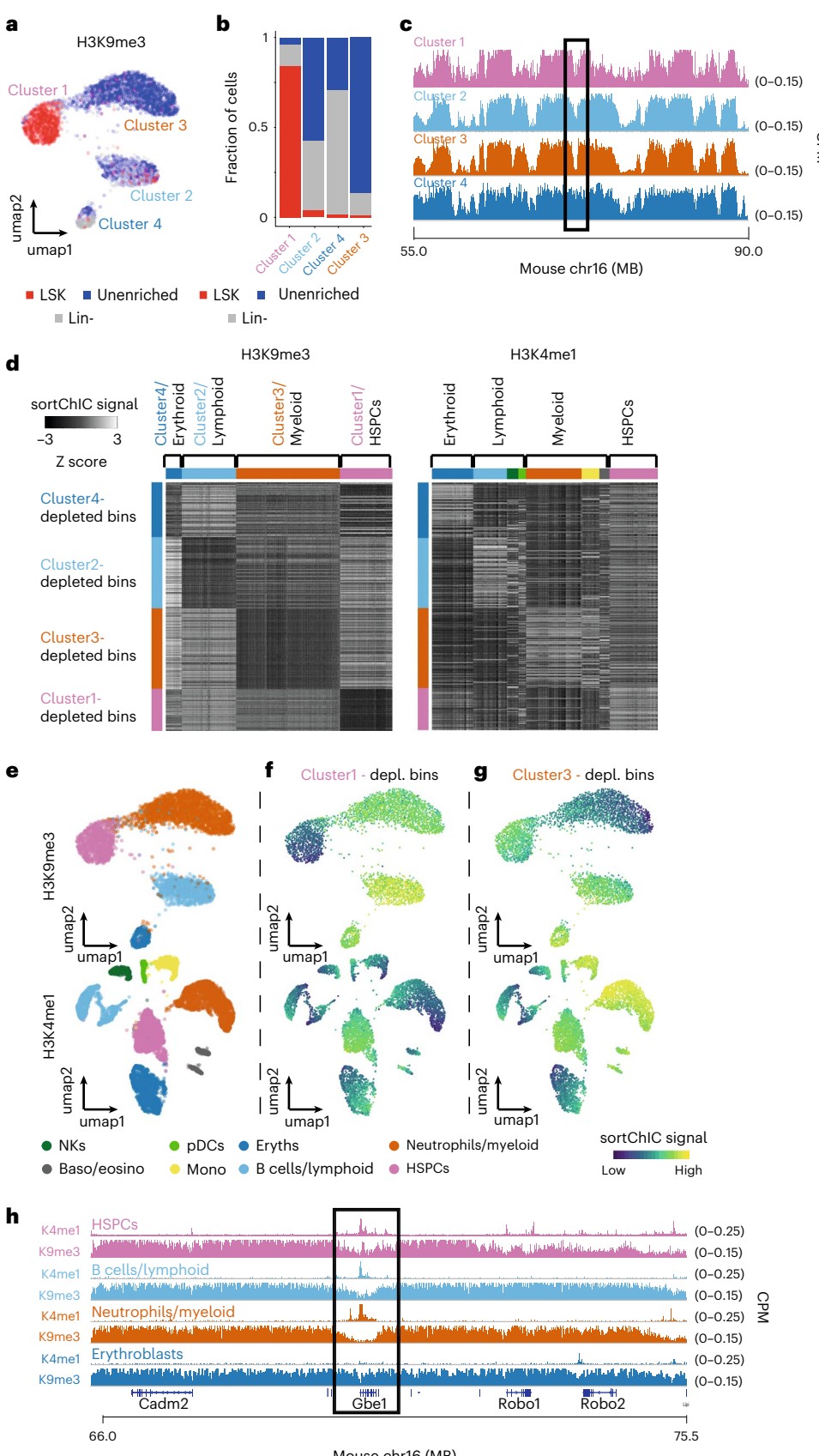

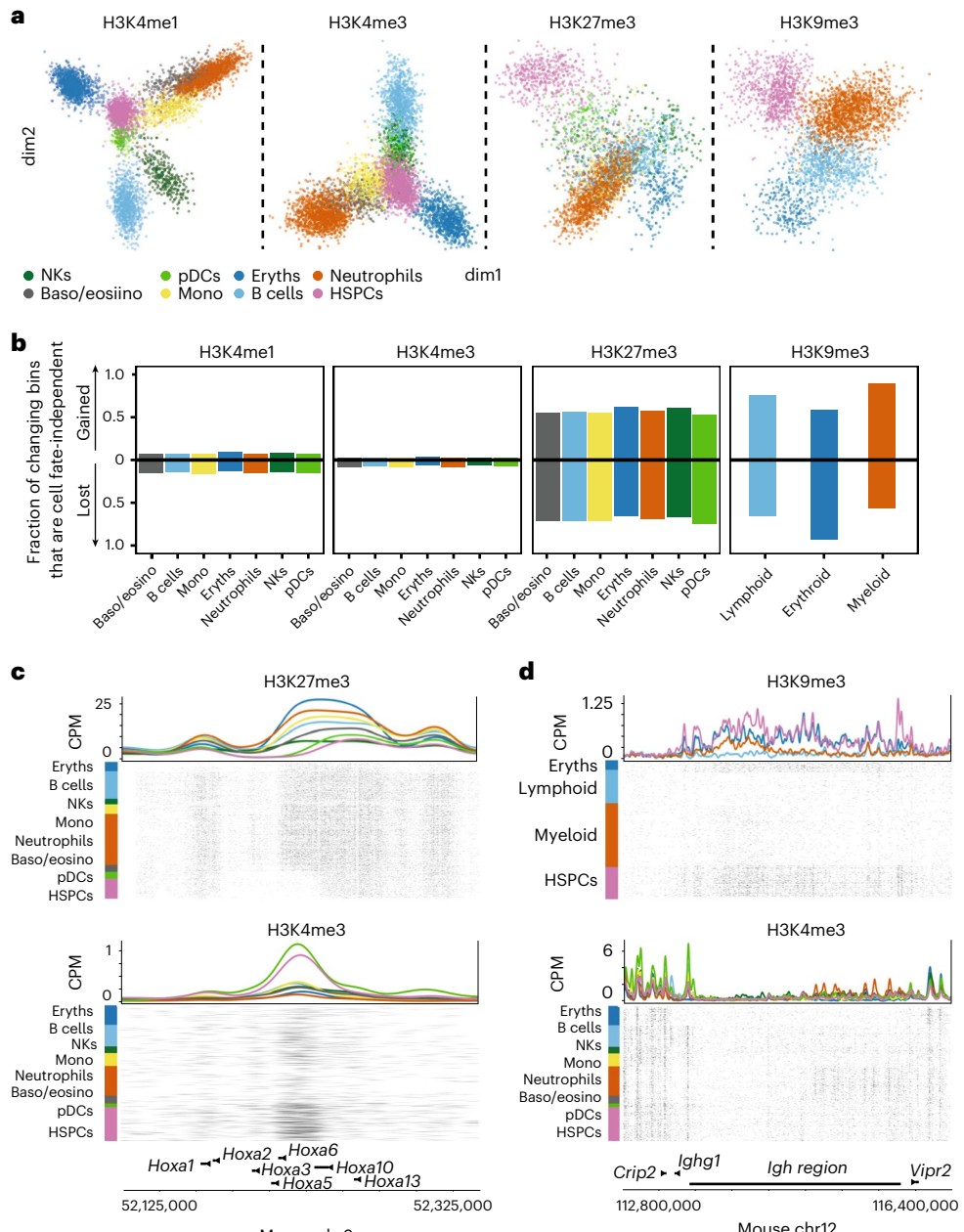

**Fig. 4 | Repressive chromatin dynamics are largely cell fate-independent. a,** Dimensionality reduction from GLMPCA (Methods) showing the two main latent factors explaining the sortChIC data for each mark. **b,** Barplot of the fraction of changing bins (Methods) that are gained or lost in all non-HSPCs relative to HSPCs. Each cell type shows two bars, one for each direction (either gained or lost). Fraction is calculated by dividing the number of bins that change cell fate independently by the number of bins that change in that cell type for that direction. **c,** Genome browser view of the *Hoxa* region showing an H3K27me3 domain that is gained during hematopoiesis. Top shows H3K27me3 and the bottom H3K4me3. **d,** Genome view of the *Igh* region displaying the loss of an H3K9me3 domain in lymphoid and myeloid cells. Top shows H3K9me3 and the bottom H3K4me3.

We used generalized principal component analysis (GLMPCA) to project the active mark data onto the two most significant axes of chromatin variation[40], which reveals a central position for HSPCs relative to other cell types, suggesting that active chromatin during hematopoiesis diverges depending on the cell type (Fig. 4a, left two panels). By contrast, clustering repressive chromatin dynamics mainly distinguishes HSPCs and differentiated cell types, (Extended Data Fig. 5a). Projecting the repressive mark data reveals a peripheral position of HSPCs compared to other cell types (Fig. 4a, right two panels). By comparing bins that gain or lose chromatin marks in mature cell types relative to HSPCs, we find more than half of the bins that gain or lose repressive marks co-occur in all other cell fates (Fig. 4b), suggesting

that changes in repressive chromatin during hematopoiesis are independent of cell fate. By contrast, only 8% of bins in active chromatin show cell-type-independent changes. Differences between HSPCs and non-HSPCs at affected bins show distinct separation between HSPCs and non-HSPCs in repressive marks. We do not observe this for active marks (Extended Data Fig. 5b), corroborating that a large fraction of changes in repressive chromatin is independent of cell fate. These cell fate-independent changes are exemplified for H3K27me3 at the *Hoxa* region, which shows low levels of H3K27me3 in HSPCs, which are upregulated in differentiated cell types (Fig. 4c). In addition, HSPCs at the immunoglobulin heavy chain (*Igh*) region carry high levels of H3K9me3, which is lost in myeloid and lymphoid cells, suggesting

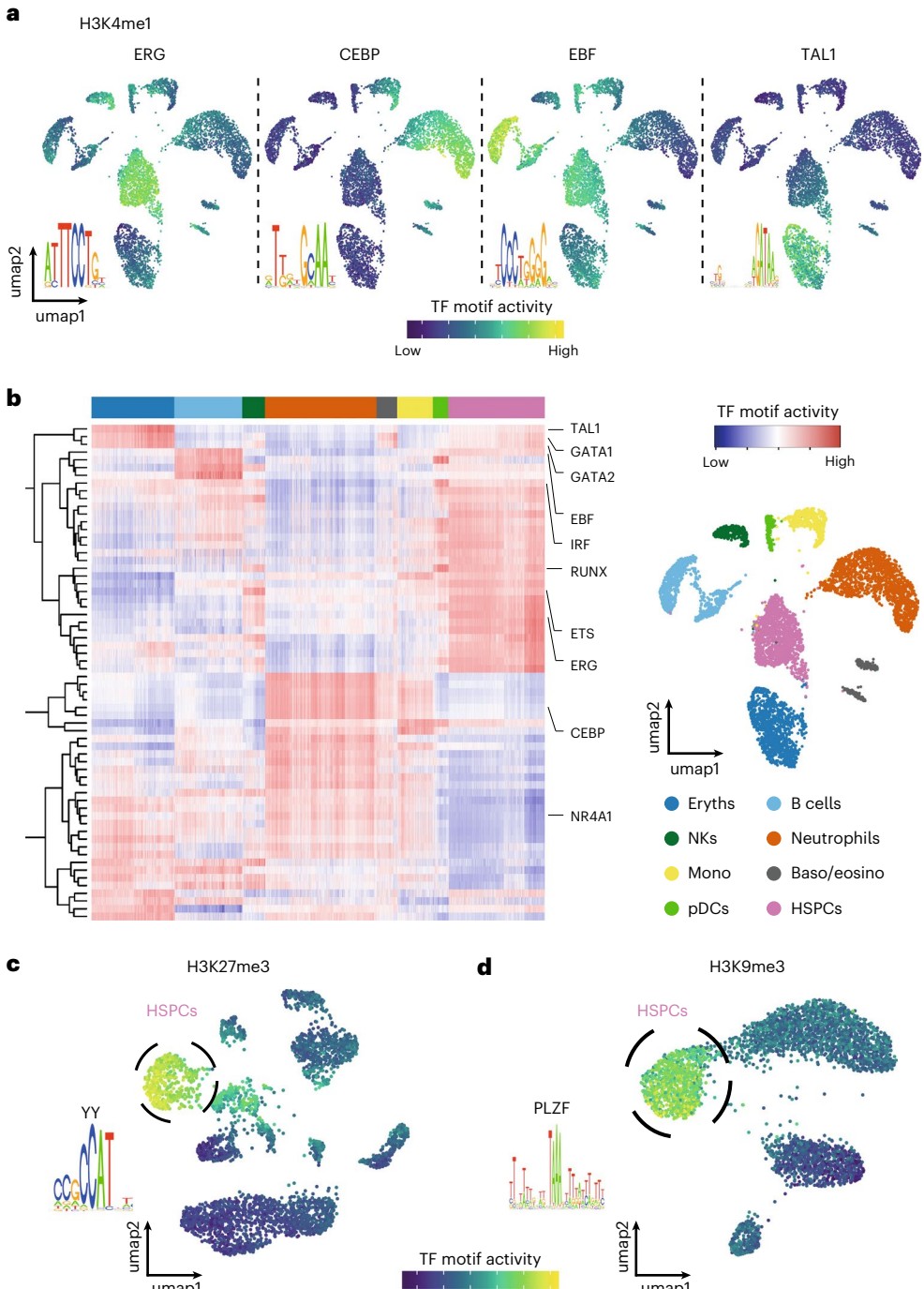

**Fig. 5 | TF motifs underlie active and repressive chromatin dynamics in hematopoiesis. a**, Examples of four TF motifs whose activities are predicted to drive cell-type-specific H3K4me1 distributions. The ERG motif is predicted to be active in HSPCs, the CEBP motif in neutrophils, the EBF motif in B cells and the TAL1 motif in erythroblasts. Cell type for each cell cluster is labeled in **b**. **b**, Heatmap of H3K4me1 TF motif activities in single cells. Rows represent motifs. Columns are individual cells whose cell types are annotated by the top color bar. The right panel shows an H3K4me1 UMAP colored by cell types, with cell-type-to-color legend below. **c**. Predicted H3K27me3 activity of a motif belonging to the Yin Yang (YY) protein family in single cells. Circled cluster is enriched for HSPCs. **d**, Predicted H3K9me3 activity of PLZF motif in single cells. Circled cluster is enriched for HSPCs.

that this region, encoding the heavy chains of immunoglobulins, is derepressed during differentiation (Fig. 4d).

Next, we ask whether H3K27me3 and H3K9me3 regulate distinct processes. We confirm that H3K27me3 dynamics occur at TSS-proximal GC-rich regions while H3K9me3 is dynamic at TSS-distal AT-rich regions (Extended Data Fig. 5c–d)[20]. Gene ontology (GO) analysis of H3K9me3 regions unique to HSPCs shows enrichment of phagocytosis, complement activation and B-cell-receptor signaling (Extended Data Fig. 5e), suggesting that HSPCs use H3K9me3 to repress genes that are required in differentiated blood cells. In contrast, GO analysis of HSPC-specific H3K27me3 regions does not show enrichment for biological processes related to blood development.

Taken together, we find that during differentiation, intermediate levels of active chromatin marks in HSPCs are up- or down-regulated

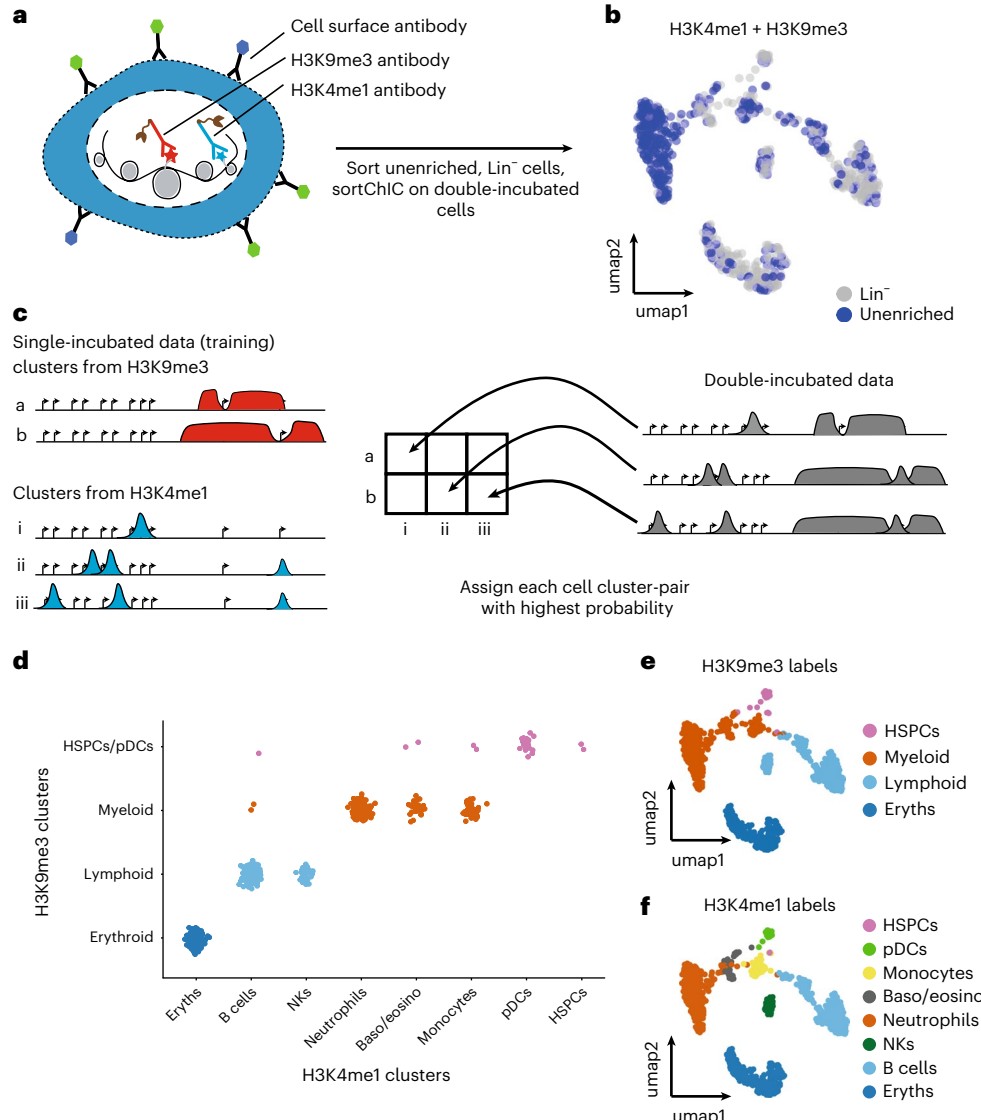

**Fig. 6 | Distinct cell types can share similar heterochromatin landscapes.**
**a**, Double-incubation experiment produces cuts associated with either
H3K4me1 or H3K9me3 (H3K4me1 + H3K9me3). **b**, UMAP representation of the
H3K4me1 + H3K9me3 landscape in unenriched and lineage-negative cells in
the BM. **c**, Schematic of how the standard single-incubated data can produce
a model of which cluster-pair (one from H3K9me3, the other from H3K4me1)
generates the observed double-incubated data. **d**, Output of cluster-pair
predictions from H3K4me1 + H3K9me3 double-incubated cells. Cells are
colored by their predicted H3K9me3 clusters. **e,f**, UMAP representation of the
H3K4me1 + H3K9me3 landscape, colored by their predicted H3K9me3 cluster (**e**)
or H3K4me1 cluster (**f**).

depending on the specific cell fate. In contrast, most dynamic
repressive chromatin regions are gained or lost independent of the
specific cell fate.

## TF motifs underlie chromatin dynamics
Next, we ask whether regulatory DNA sequences underlying the
sortChIC data can explain the chromatin changes. We hypothesize that
regions with correlated sortChIC signal across cells can be explained
by TF binding motifs shared across these regions[41,42] (Extended Data
Fig. 6a). We adapted MARA, a ridge regression framework, to infer TF
motif activities in single cells. SortChIC signals are the observed vari-
ables, TF binding motifs are covariates and TF motif activities are latent
variables to be inferred. We find statistically significant TF motifs that
explain correlations in single-cell chromatin dynamics across different
genomic regions. We use TF motif activity[42–46] as a term to connect our
method to earlier contributions to this problem. Overlaying the pre-
dicted single-cell TF motif activities onto the UMAP shows the expected

cell-type-specific TF motif activities. We find high ERG motif activity in
HSPCs[47] (Fig. 5a, left), high CEBP motif activity in neutrophils[48,49] (Fig.
5a, mid-left), high EBF motif activity in B cells[50] (Fig. 5a, mid-right) and
high TAL1 motif activity in erythroblasts[51] (Fig. 5a, right), in agreement
with the reported role of each TF.

We summarize the inferred single-cell TF activities underlying
the cell-type-specific distribution of active H3K4me1 in Fig. 5b. We
predict motifs active in pDCs belonging to the IRF and RUNX family
(Fig. 5b and Extended Data Fig. 6b–d), consistent with their role in type
1 interferon secretion[52,53], dendritic cell progenitor development[54] and
pDC migration[55], respectively. We find natural killer (NK) cells to have
high E26 transformation-specific (ETS) family motif activity (Fig. 5b and
Extended Data Fig. 6b,e), consistent with the role *of Ets1* in the develop-
ment of natural killer and innate lymphocyte cells[56,57]. Finally, we predict
TFs that have the lowest activity in HSPCs and pDCs, such as the NR4A
family (Fig. 5b and Extended Data Fig. 6b,f). Considering that NR4A
family members are highest expressed in HPSCs (data not shown), we

conclude that NR4A mainly prevents enhancer activation, consistent with a repressive function of *Nr4a1* in HSPCs[58,59]. The low activity of several TFs suggests that pDCs could be in a more progenitor-like state, consistent with our pseudobulk clustering results in H3K4me1, H3K4me3 and H3K27me3 (Extended Data Fig. 5a).

We apply our TF motif analysis to the two repressive chromatin landscapes to predict motifs that explain HSPC-specific distributions. In H3K27me3, we predict a CCAT motif belonging to the Yin Yang family[60], specifically active in HSPCs (Fig. 5c). The *Yy1* gene encodes a polycomb group protein, shown to regulate HSC self-renewal[61]. In H3K9me3, we predict an AT-rich motif belonging to the transcriptional repressor PLZF, specifically active in HSPCs (Fig. 5d), that has been implicated in regulating the cell cycle of HSCs[62].

Taken together, our framework predicts TFs underlying cell-type-specific chromatin dynamics. We suggest that differentiating cells decide which active regions to up- or down-regulate depending on the cell-type-specific TFs that associate with these regions.

## Distinct cell types can share similar heterochromatin states

To understand the relationship between the eight cell types identified by histone marks of gene-rich regions (H3K4me1, H3K4me3 and H3K27me3) to the four clusters identified by H3K9me3, we stain cells with both H3K4me1 and H3K9me3 antibodies[63]. This double-incubation strategy generates cuts that come from both H3K4me1 and H3K9me3, and uses our single mark sortChIC data to infer the relationships between the two marks in single cells (Fig. 6a). We sort Lin⁻ and unenriched cells to profile abundant and rare cell types. Joint UMAP landscapes reveal clusters that are depleted or enriched for mature lineage markers (Fig. 6b). We use clusters from H3K4me1 and H3K9me3 single-incubated data to develop a model of how the double-incubated data could be generated (Fig. 6c).

For this, we select 811 regions associated with cell-type-specific genes found in our H3K4me1 analysis (Fig. 2e) and 6,085 cluster-specific regions (50 kb bins) found in our H3K9me3 analysis (Extended Data Fig. 5a, right panel) as features in our model, making a total of 6,896 regions. We verify that these features show cluster-specific differences, by clustering the single-incubated H3K4me1 and H3K9me3 signal across cell types (Extended Data Fig. 7a,b).

Because we do not know which cluster from H3K4me1 pairs with which cluster from H3K9me3, we generate an in-silico model of all possible pairings (Fig. 6c, left). For each double-incubated cell, we perform model selection to select the cell pair with the highest probability (Fig. 6c, right, and Extended Data Fig. 7c–e). This selection reveals that cell types share a common heterochromatin landscape, reflecting their myeloid[64] or lymphoid lineage[65] (Fig. 6d). Erythroblasts do not share a heterochromatin landscape with any other cell type. Surprisingly, we find pDCs associated with the HSPC-enriched H3K9me3 landscape, suggesting that these cells may have already committed toward a pDC fate through active chromatin, while their heterochromatin remains undifferentiated.

This confirms that distinct cell types in related lineages can share their heterochromatin state (Fig. 6e,f), suggesting a hierarchical model where changes in heterochromatin might restrict lineages and changes in active chromatin define cell types within lineages.

## Distinct repressive chromatin trajectories in hematopoiesis

To systematically analyze a continuous trajectory from fluorescence-activated cell sorting (FACS)-validated HSCs to differentiated cell types across histone modifications, we expand our dataset to include different HSPC subpopulations and cKit⁺ progenitor cells. Specifically, we sort HSCs, including both long-term (LT) and short-term (ST) HSCs, MPPs, common myeloid progenitors (CMPs), and megakaryocyte/erythrocyte progenitors (MEPs). Furthermore, we validate our differentiated cell types by sorting B cells, NK cells, erythroblasts, neutrophils, monocytes, pDCs and cDCs (Extended Data Fig. 8a). In total, we increase our BM dataset by 17,270 new cells across H3K4me1, H3K4me3, H3K27me3 and H3K9me3 (Extended Data Fig. 8b), giving a total of 39,857 cells in our dataset.

A subset of the new sortChIC cells has combinations of Sca1, cKit and Lin marker levels from FACS that allow the definition of a FACS-based differentiation stage (Fig. 7a). We plot these Sca1, cKit, Lin-stained cells onto a ternary plot to project cells along a FACS-defined differentiation trajectory. Cells arrange along a continuum of differentiation potential as follows: from uncommitted progenitors (Sca1⁺, cKit⁺ and Lin⁻) and committed progenitors (Sca1⁻, cKit⁺ and Lin⁻) to mature cells (Sca1⁻, cKit⁻ and Lin⁺). Plotting relative levels of Sca1, cKit and Lin onto the UMAP reveals HSCs, progenitors and mature cells (Fig. 7b).

Next, we use the labeled cells from FACS (Extended Data Fig. 8a) to assign each cell to a cell type in a supervised and probabilistic manner (Extended Data Fig. 9a–e), creating a high-confidence dataset of 14 subtypes (Fig. 7c). Of note, we find that monocytes are epigenetically distinct from neutrophils and DCs in H3K4me1, H3K4me3 and H3K27me3, but in H3K9me3 all mature myeloid cell types appear to cluster together (Fig. 7c and Extended Data Fig. 9a–c). We validate the presence of pDCs in our dataset, which forms distinct islands in H3K4me1, H3K4me3 and H3K27me3 but are spread across the HSPC cluster in H3K9me3 (Extended Data Fig. 9b).

We analyze neutrophil, B cell, erythroblast and HSPC-specific marker gene sets (±5 kb around TSS) for H3K4me1, H3K4me3 and H3K27me3 alterations from HSCs to different mature cell types. For mature cell-type-specific genes, we find that active marks start with intermediate levels in HSCs, which diverge during differentiation into mature cell types (Fig. 7d and Extended Data Fig. 10a–c). In contrast, marker genes of mature cell types show low H3K27me3 in HSCs that increase during differentiation in cell types that do not express them (Fig. 7d and Extended Data Fig. 10b–c, right). Genes specifically expressed in HSPCs lose active marks and accumulate H3K27me3 in all differentiation trajectories (Extended Data Fig. 10d).

To summarize these trajectory dynamics, we take dynamic bins (Supplementary Table 1) and apply principal component analysis (PCA) (Fig. 7e). To estimate chromatin velocities for each mark, we fit a trajectory-specific cubic spline across pseudotime for each bin, then calculate the derivatives with respect to pseudotime. Bin-level velocities are then projected onto the PCA for each histone mark (Fig. 7e). In active marks, we find trajectories that diverge according to erythroid, myeloid and lymphoid lineages. Repressive chromatin, by contrast, shows cell-type-independent changes before lineage specification. At the bin level, we use regions that are upregulated for each histone mark independently for neutrophils, B cells or erythroblasts

**Fig. 7 | Trajectory analysis across stem, progenitor and mature cell types reveal histone mark-specific chromatin velocities. a**, sortChIC design to capture stem, progenitor and mature cell types during hematopoiesis. Ternary plot of cells for Sca1, cKit and Lin marker levels measured by FACS. **b**, Sca1, cKit and Lin-stained cells plotted in UMAP space. Cells with staining are colored according to their relative levels of Sca1, cKit and Lin, as coded in **a**. Cells unstained for these surface molecules are colored gray. **c**, UMAP integrating all BM sortChIC data for each of the four histone modifications. Cell type identity is based on the sorted cell types explained in Extended Fig. 8a (number of cells for H3K4me3, *n* = 10,952; H3K4me1, *n* = 12,085; H3K27me3, *n* = 7,934 and H3K9me3,

*n* = 8,886). **d**, Mean sortChIC signal of neutrophil marker genes (defined from heatmap Fig. 2e). The same 150 regions are used for each histone modification. **e**, First two principal components for the sortChIC data. Chromatin velocities are calculated for each bin and then projected onto the PCA for each modification separately (Methods). **f**, Mean sortChIC signal for bins that are upregulated in neutrophils relative to HSPCs across cell types for the four histone marks independently. Regions are defined for each histone modification separately (H3K4me3, 1,009 bins; H3K4me1, 4,473 bins; H3K27me3, 2,549 bins and H3K9me3, 2,838 bins). Density plots below show the distribution of cell types along the neutrophil trajectory (HSCs, LTs, STs, MPPs, CMPs and neutrophils).

relative to HSPCs and plot the mean histone mark levels per cell along pseudotime (Fig. 7f, Supplementary Fig. 1a–b, regions defined previously, and Supplementary Table 1). For all three bin sets, we find that

active marks diverge across cell types, while repressive marks show dynamics that are shared across cell types consistent with our earlier findings (Fig. 4b).

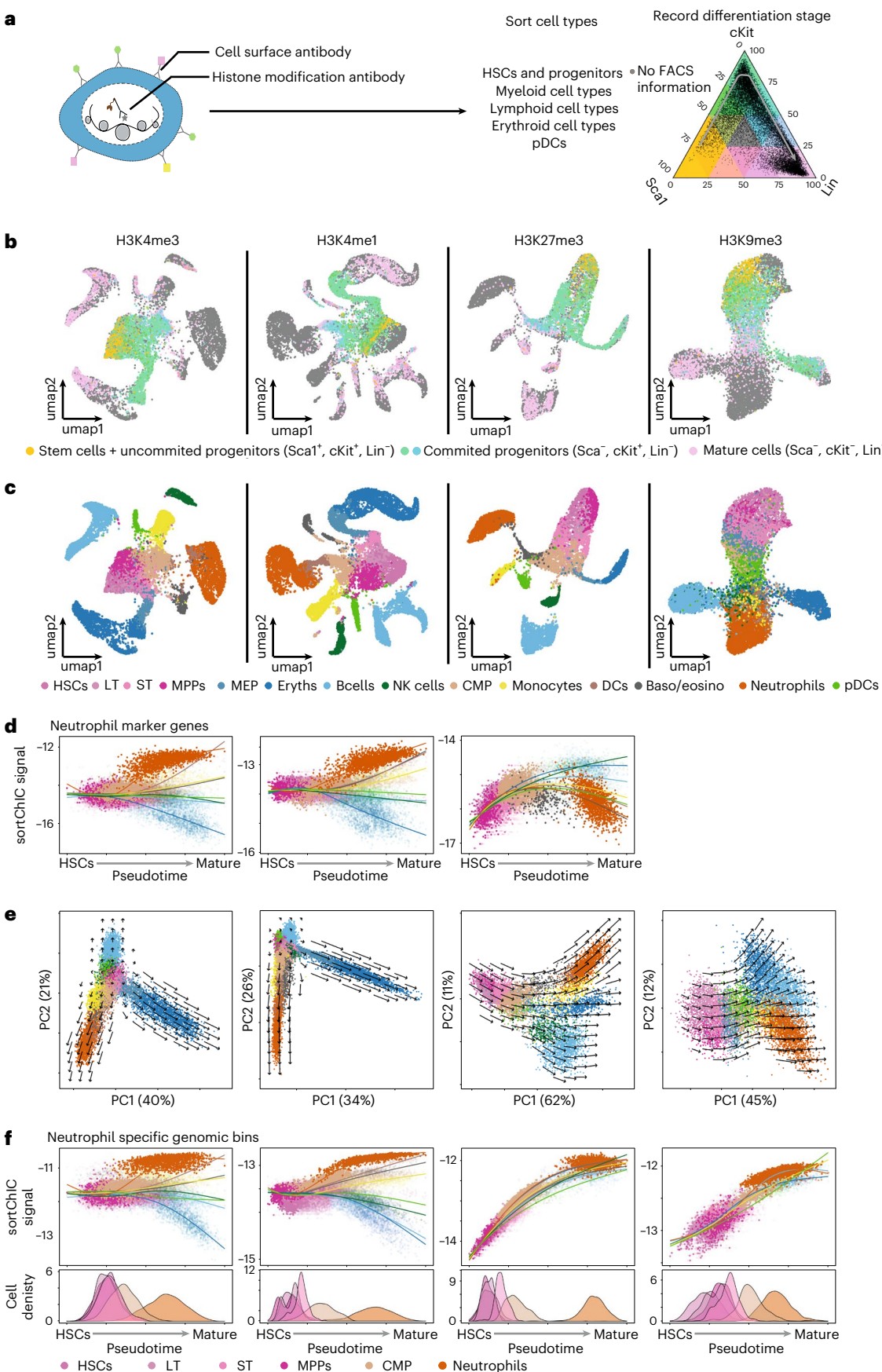

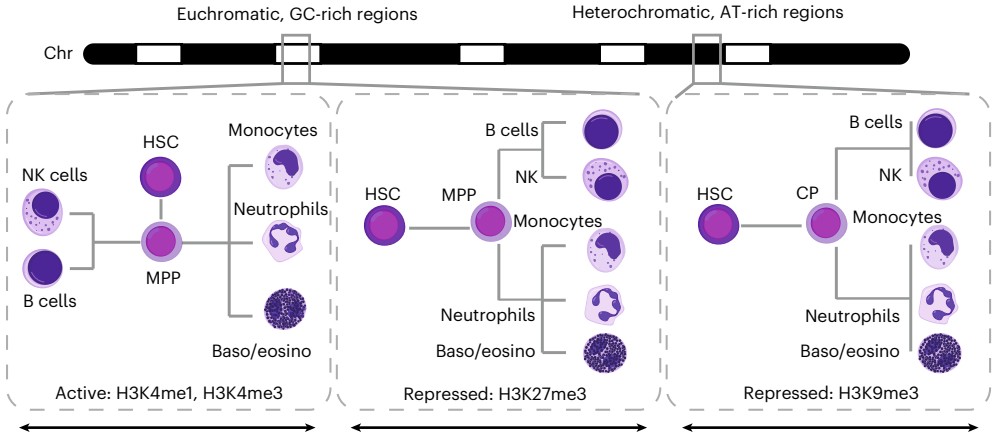

**Fig. 8 | Hierarchical chromatin regulation during blood formation.** Graphical summary of chromatin dynamics as dendrograms showing relationships between HSPCs and differentiated cells. During hematopoiesis, the direction of change in active chromatin depends on the specific cell fate, resulting in global differences that are larger between differentiated cell types from different lineages. By contrast, many regions gain or lose repressive marks during hematopoiesis independent of the specific cell fate, resulting in global differences that are the largest between HSPCs and differentiated cell types. Dynamics in active marks and H3K27me3-marked repressive chromatin reveal cell type information, while dynamics in heterochromatin regions marked by H3K9me3 reveal lineage information. CP, common progenitors.

## Chromatin commitment coincides with lineage restriction

To compare the global dynamics of the four different histone marks along a common trajectory, we use the marker levels of Sca1, cKit and Lin and asked when global chromatin states are specified along the Sca1-cKit-Lin trajectory. Overlaying the relative levels of Sca1, cKit and Lin onto the PCA shows that Sca1 levels are already low when chromatin has specified the myeloid (CMPs) or erythroid lineage (MEPs; Supplementary Fig. 2a). Plotting principal component 1 along the Sca1-cKit-Lin trajectory shows that first differences on chromatin level can be observed at the exit of multipotency, when MEPs and CMPs emerge after the loss of Sca1 (Supplementary Fig. 2b,c), suggesting that chromatin changes co-occur with lineage commitment. These results are in line with previous studies identifying a switch from multilineage priming to lineage restriction on marker genes during progenitor cell commitment[66]. Overall, we apply sortChIC to interrogate FACS-validated rare subpopulations and differentiated cell types in the BM, enabling systematic analysis of active and repressive chromatin dynamics during hematopoiesis.

## Discussion

Here we provide a comprehensive map of chromatin regulation at both euchromatic and heterochromatic regions during blood formation. We find that repressive chromatin shows distinct dynamics compared with active chromatin, demonstrating that profiling repressive chromatin regulation in single cells reveals new dynamics. Active chromatin premarks in HSPCs genes of all lineages and is up- or down-regulated depending on the specific cell fate, mediated by cell-type-specific TFs. Consequently, active chromatin shows divergent changes for different blood cell fates (Fig. 8, left panel). In contrast, changes in repressive chromatin often occur in the same direction regardless of the specific cell fate, resulting in large differences between HSPCs and mature cell types (Fig. 8, middle and right panel). In accordance with the premarked active chromatin state in HSCs, the majority of mature cell-type-specific genes show low levels of H3K27me3 in HSCs and consolidate their differentiation choice by silencing genes specific to HSCs and of the unchosen trajectory. This progressive transition to a restricted chromatin state agrees with previous studies showing a genome-wide transition during ES cell differentiation[67]. Although our results are correlative, previous work characterizing the consequences of HSC-specific deletion of EED[68], a core component of both PRC1 and

PRC2, showed a loss of differentiation capacity, while preserving HSCs self-renewal. This suggests an integral role of H3K27me3 after the onset of lineage commitment in hematopoiesis.

Our findings further expand the role of H3K9me3[16]. We find that H3K9me3 changes underlie the lineage restriction in hematopoiesis and are rewired as HSPCs differentiate. Although in vivo dynamics in H3K9me3 have been reported during early development[16–18], our results extend the knowledge of H3K9me3 dynamics to homeostatic renewal in adult physiology. Joint analysis of active and repressive marks corroborates the hierarchical chromatin changes and shows a similarity between pDCs and HSPCs[69,70] in their heterochromatin state.

Our FACS sorting strategy profiled the epigenomes of rare and abundant cell types in the BM. Although our analysis did not find clear subpopulations within rare progenitor cells previously observed in scRNA-seq studies[4,71], the cell type resolution obtained with sortChIC is comparable to scRNA-seq studies. Rather than a way to further subcategorize existing cell types, sortChIC profiles layers of regulation that guide differentiation. If the sensitivity can be further improved, additional chromatin states might become visible that are indistinguishable from scRNA-seq. Future multi-omics studies integrating the detection of chromatin modifications with transcription[72–74] should further facilitate the integrated analysis of diverse histone modifications and allow us to more clearly understand how these multiple layers of gene regulation are related.

## Online content

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

## Methods

Our research complies with all relevant ethical guidelines. Experimental procedures were approved by the Dier Experimenten Commissie of the Royal Netherlands Academy of Arts and Sciences and performed according to the guidelines.

### Animal experiments

Primary BM cells were collected from 3-month-old male C57BL/6 mice. Femur and tibia were extracted, and the bones ends were cut away to access the BM, which was flushed out using a 22 G syringe with HBSS (-Ca, -Mg, -phenol red; Gibco, 14175053) supplemented with Pen-Strep and 1% FCS. The BM was dissociated and debris was removed by passing it through a 70 µm cell strainer (Corning, 431,751). Cells were washed with 25 ml supplemented HBSS before lineage marker staining was performed following the instructions of the EasySep Mice Hematopoietic Progenitor Cell Isolation Kit (Stemcell), using half of the recommended concentration of the biotinylated antibodies. This was followed by 30 min incubation at 4 °C with a staining layout-dependent antibody cocktail detailed below. Where indicated lineage depletion was performed by incubating cells with magnetic streptavidin beads following instructions of the EasySep Mice Hematopoietic Progenitor Cell Isolation Kit. After two additional washes with HBBS (+PS, +FCS), cells were prepared following the sortChIC protocol for the four different histone modifications.

### Cell culture

K562 cells (ATCC CCL-243) were grown in RPMI 1640 Medium GlutaMAX, supplemented with 10% FCS, Pen-Strep and nonessential amino acids. After collecting, cells were washed three times with room temperature PBS before continuing with the sortChIC protocol.

### sortChIC-seq: Cell preparation: fixation.

Three buffers are used for the majority of cell preparation. A basic ChIC buffer (47.5 ml H2O RNAse free, 1 ml 1 M HEPES pH 7.5 (Invitrogen), 1.5 ml 5 M NaCl, 3.6 µl pure spermidine solution (Sigma Aldrich), 0.05% Tween20), a Wash buffer (Basic ChIC buffer with 1 Ethylenediaminetetraacetic acid (EDTA)-free protease inhibitor cocktail tablet per 50 ml (Sigma Aldrich)) as well as a Antibody incubation buffer (Wash buffer with 4 ml ml−1 0.5 M EDTA). All steps performed on ice were as follows: in step 1, cells were resuspended in 300 µl PBS per 1 million cells in a 15 ml protein low binding falcon tube and 700 µl ethanol (−20 °C precooled) per 1 million cells are added while vertexing cells at middle speed. In step 2, cells were fixed for 1 h at −20 °C. In step 3, after fixation, cells were washed twice in 1 ml antibody incubation buffer. In case cells had to be stored before sorting, DMSO was added to a final concentration of 10% and cells were frozen at −80 °C. After thawing, cells are washed once in 0.5 ml antibody incubation buffer before continuing with pA-MN targeting.

### sortChIC-seq: Cell preparation: nuclei.

Cells were washed once in 1 ml antibody incubation buffer (0.05% Tween replaced by 0.05% Saponin for this and following steps with nuclei). Nuclei were isolated by further Saponin incubation overnight in parallel to the antibody staining. For BM, we sorted nine plates each for H3K4me1, H3K4me3 and H3K9me3.

### sortChIC-seq: pA-MN targeting.

In step 4, cells were pelleted at 500 g for 4 min and resuspended in 200 µl antibody incubation buffer per 1 million cells and were aliquoted into 0.5 ml protein low binding tubes containing the primary histone mark antibody (details can be found in the Supplementary Note section Materials section) diluted in 200 µl antibody incubation buffer; in step 5, cells were incubated overnight at 4 °C on a roller, (step 6) before they were washed once with 500 µl Wash Buffer. In the case of double-labeling experiments, cells were incubated with antibodies against H3K4me1 and H3K9me3 together at the same concentrations as for the single-mark experiments. Afterwards (step 7), cells were resuspended in 500 µl wash buffer containing pA-MN

(3 ng ml−1) and Hoechst 34580 (5 µg ml−1) and (step 8) incubated for 1 h at 4 °C on a roller. In step 9, finally, cells were washed an additional two times with 500 µl Wash Buffer before passing them through a 70 µm cell strainer (Corning, 431751).

### sortChIC-seq: FACS sorting.

In step 10, for all experiments, cells were gated additionally to cell surface markers for G1 cell cycle stage based on the Hoechst staining on an Influx FACS machine into 384 well plates, containing 5 µl sterile filtered mineral oil (Sigma Aldrich) per well, using forward scatter and trigger pulse width to further remove cell doublets. Cells were sorted using index sorting, which records FACS information for every sorted well. To further exclude missorting of more than the intended cell, we used custom sort settings−objective: single, number of drops=1, extra coincidence=complete empty (no signal in the previous and next drop) and phase mask=center 10/16 (cell is in the middle of the sorted drop).

Sort layouts for separate experiments can be found in Extended Data Figs. 1a, 3a and 8a, with total number of plates sorted per condition found in Supplementary Table 4. Antibody details can be found in the Supplementary Note section Materials section. Data was collected using BD FACS software (version 1.2.0.124).

### sortChIC-seq: pA-MN activation.

The following small volumes were distributed using a Nanodrop II system (Innovadyme) and plates were spun for 2 min at 4 °C and 2,000g after each reagent addition.

In step 11, 100 nl of basic ChIC buffer, containing 2 mM CaCl2, was added per well to induce pA-MN mediated chromatin digestion. In step 12, for digestion, plates were incubated for 30 min in a PCR machine set at 4 °C. Afterwards (step 13), the reaction was stopped by adding 100 nl of a stop solution containing 40 mM EGTA (chelates Ca2+ and stops MN, Thermo, 15425795), 1.5% NP40 and 10 nl 2 mg ml−1 proteinase K (Invitrogen, AM2548). In step 14, plates were incubated in a PCR machine for further 20 min at 4 °C, before chromatin is released and pA-MN was permanently destroyed by proteinase K digestion at 65 °C for 6 h followed by 80 °C for 20 min to heat inactivate proteinase K. Afterwards, plates can be stored at −80 °C until further processing.

### sortChIC-seq: Library preparation.

In step 15, DNA fragments are blunt-ended by adding 150 nl end repair mix (Supplementary Table 5) per well and incubating for 30 min at 37 °C followed by 20 min at 75 °C for enzyme inactivation. In step 16, blunt fragments are subsequently A-tailed by adding 150 nl per well of A-tailing mix (Supplementary Table 6) and incubating for 15 min at 72 °C. Through AmpliTaq 360's strong preference to incorporate dATP as a single base overhang even in the presence of other nucleotides, a general dNTP removal is not necessary.

Next fragments are ligated to T-tail containing forked adapters (see Supplementary Note section Materials for sequences).

In step 17, for ligation, 50 nl of 5 µM adapter in 50 mM Tris pH 7 is added to each well with a mosquito HTS (ttplabtech). After centrifugation (step 18), 150 nl of adapter ligation mix (Supplementary Table 7) are added before (step 19) plates are incubated for 20 min at 4 °C, followed by 16 h at 16 °C for ligation and 10 min at 65 °C to inactivate ligase.

In step 20, before pooling 1 µl of Nuclease-free water was added to each well to minimize material loss. In step 21, ligation products were pooled by centrifugation into oil-coated VBLOK200 Reservoir (ClickBio) at 500g for 2 min and (step 22) the liquid face was transferred into 1.5 ml Eppendorf tubes and (step 23) was purified by centrifugation at 13,000g for 1 min and transfer into a fresh tube twice. In step 24, DNA fragments were purified using Ampure XP beads (Beckman Coulter−prediluted 1 in 8 in bead binding buffer−1 M NaCl, 20% PEG8000, 20 mM Tris, pH = 8, 1 mM EDTA) at a bead-to-sample ratio of 0.8. In step 25, after 15 min incubation at room temperature, beads were washed twice with 1 ml 80% ethanol resuspending the beads during the first wash and (step 26) resuspended in 8 µl nuclease-free water.

After 2 min elution, the supernatant was (step 27) transferred into a fresh 0.5 ml tube. In step 28, the cleaned DNA is then linearly amplified by in vitro transcription adding 12 μl of MEGAscript T7 Transcription Kit (Thermo Fisher Scientific, AMB13345) for 12 h at 37 °C. In step 29, template DNA is removed by the addition of 2 μl TurboDNAse (IVT kit) and incubation for 15 min at 37 °C. In step 30, the produced RNA is further purified using RNA Clean XP beads (Beckman Coulter) at 0.8 beads to sample ratio and samples are resuspended in 22 μl of Nuclease-free water. In step 31, RNA is fragmented by mixing in 4,4 μl fragmentation buffer (200 mM Tris-acetate pH 8.1, 500 mM KOAc, 150 mM MgOAc) and incubation for 2 min at 94 °C. In step 32, fragmentation is stopped by transferring samples to ice, adding 2.64 μl 0.5 M EDTA and another bead cleanup and samples are resuspended in 12 μl nuclease-free water.

In step 33, 5 μl of the RNA is primed for reverse transcription by adding 0.5 μl dNTPs (10 mM) and 1 μl random hexamer reverse transcription primer 20 μM (for sequence see Supplementary Note section Materials) and (step 34) hybridizing it by incubation at 65 °C for 5 min followed by direct cool down on ice. In step 35, reverse transcription is performed by further addition of 2 μl first strand buffer (part of Invitrogen, 18064014), 1 μl DTT 0.1 M (Invitrogen, 15846582), 0.5 μl RNAseOUT (Invitrogen, LS10777019) and 0.5 μl SuperscriptII (Invitrogen, 18064014) and (step 36) incubating the mixture at 25 °C for 10 min followed by 1 h at 42 °C. In step 37, single-stranded DNA is purified through incubation with 0.5 μl RNAse A (Thermo Fisher Scientific, EN0531) and (step 38) incubation for 30 min at 37 °C. In step 39, a final PCR amplification to add the Illumina small RNA barcodes and handles is performed by adding 25 μl of NEBNext Ultra II Q5 Master Mix (NEB, M0492L), 11 μl nuclease-free water and 2 μl of RP1 and RPIx primers (10 μM).

In step 40, PCR is performed with following protocol, activation for 30 s at 98 C, 8–12 cycles (depending on starting material) 10 s at 98 C, 30 s at 60 C, 30 s at 72 °C, final amplification 10 min at 72 °C (step 41) PCR products are cleaned by two consecutive DNA bead clean-ups with a 0.8X bead-to-sample ratio. In step 42, the final product was eluted in 7 μl nuclease-free water, and the abundance and quality of the final library are assessed by QUBIT and bioanalyzer.

## pA-MN production
The pA-MN fusion protein was produced following the methods section in ref.[24] (details can be found in Supplementary Note section Materials).

## Statistics and reproducibility
No statistical method was used to predetermine the sample size. Low-quality cells (for example, number of cuts below threshold, cuts not containing expected MN cut motif, and cells with unspecific cuts) were removed from further analysis. The experiments were not randomized. The investigators were not blinded to allocate during experiments and outcome assessment.

## Data preprocessing
We developed a preprocessing pipeline called SingleCellMultiOmics (version v.0.1.25) to process sortChIC data (https://github.com/BuysDB/SingleCellMultiOmics/wiki). The pipeline for sortChIC processes raw fastq files through the following software:

Demultiplexing is performed with demux.py (from SCMO v0.1.25) and adaptors are trimmed using cutadapt (version 3.5). Reads are mapped with bwa (version: 0.7.17-r1188) and are assigned to molecules with bamtagmultiome.py (SCMO v0.1.25). Finally, count tables are generated using bamToCountTable.py (SCMO v0.1.25). The code was run using python version 3.7.6 and R version 4.1.2. Details can be found in the Supplementary Note section Methods.

An example of this full pipeline is available in the sortchicAnalysis git repository: https://github.com/jakeyeung/sortchicAnalysis/tree/main/example_processing_pipeline.

## Calculating reads falling in peaks in sortChIC for K562 cells
For each histone modification, we merged K562 single-cell sortChIC data, and used the resulting pseudobulk as input for *hiddenDomains*[75], with minimum peak length of 1,000 bp. We determined 40,574, 58,257, 28,499 and 28,380 peaks for H3K4me1, H3K4me3, H3K27me3 and H3K9me3, respectively. For each histone modification, we counted the fraction of total reads that fall within each set of peaks.

## Comparison of sortChIC data with other single-cell chromatin profiling assays
To perform a fair comparison of sortChIC data with other similar assays, we downloaded the raw data from Bartosovic et al. (GSE163532)[32], Grosselin et al. (GSE117309)[22], Ku et al. (GSE105012)[27], Wu et al. (GSE139857)[31], Kaya-Okur et al. (GSE124557)[30] and Ku et al. (GSE139857)[28], from GEO, and mapped and quantified them using the pipelines described by the authors in the original study. For details of study-specific processing, see Supplementary Note section Materials.

## Dimensionality reduction based on multinomial models
We counted the number of cuts mapped to peaks across cells and applied the LDA model[39] (from topicmodels version 0.2–12), which is a matrix factorization method that models discrete counts across predefined regions as a hierarchical multinomial model. LDA can be thought of as a discrete version of probabilistic PCA, replacing the Gaussian likelihood with a multinomial one[76,77]. Details can be found in Supplementary Note section Materials.

## Defining eight sets of blood cell-type-specific genes for cell typing
We used the LDA outputs to define topics associated with each cell type. Details can be found in Supplementary Note section Materials.

## Defining genomic regions for dimensionality reduction
We initially defined regions based on 50 kb nonoverlapping windows genome-wide, applying LDA and using the Louvain method to define clusters to merge single-cell bam files. These merged bam files were then used to call substantially marked regions using *hiddenDomains*[75] with minimum bin size of 1 kb. We merged the regions across clusters and generated a new count matrix using the *hiddenDomains* peaks as features. This new count matrix was used as input for dimensionality reduction.

## Batch correction in dimensionality reduction
Initial LDA of the count matrix revealed batch effects in H3K4me1 and H3K9me3 between cell types of plates that contained only one sorted type. We fit a linear model in the latent space learned from LDA with a cell-type-specific batch effect to correct batch effects. Details can be found in Supplementary Note section Methods.

## Differential histone mark levels analysis
To calculate the fold change in histone mark levels at a genomic region between a cell type versus HSPCs, we modeled the discrete counts $Y$ across cells as a Poisson regression. We fitted a null model, which is independent of cell type, and a full model, which depends on the cell type and compared their deviances to predict whether a region was 'un-changing' or 'dynamic' across cell types. We implemented the model in R using glm(), details can be found in Supplementary Note section Materials.

## Defining bins above background levels for each mark
For each mark, we counted fragments falling in 50 kb bins summed across all cells. We then plotted this vector of summed counts as a histogram in log scale, which shows a bimodal distribution. We manually defined a cut-off for each mark as a background level and took bins that were above this cut-off. This cut-off resulted in 22,067, 12,661, 18,512 and 19,881 bins for H3K4me1, H3K4me3, H3K27me3 and H3K9me3, respectively.

## Calculating bins that change independent of cell type

We used a cut-off of $q < 10^{-50}$ for H3K4me1, H3K4me3 and H3K27me3, and $q < 10^{-9}$ for H3K9me3 from the deviance test statistic (details of 'differential histone mark analysis' can be found in Supplementary Note section Materials) to define bins that are changing between cell types. Details can be found in Supplementary Note section Materials.

## Predicting activities of TFs in single cells

We adapted motif activity response analysis (MARA) described in ref. [42] to accommodate the sortChIC data. Briefly, we model the log-imputed sortChIC-seq signal learned from LDA as a linear combination of TF binding sites and activities of TF motifs using a ridge regression framework:

$$\tilde{Y}_{g,c} = \sum_{m=1}^{M} N_{g,m} A_{m,c} + \epsilon$$

where $\tilde{Y}_{g,c}$ is the batch-corrected sortChIC-seq signal in genomic region $g$ in cell $c$; $N_{g,m}$ is the number of TF binding sites in region $g$ for TF motif $m$; $A_{m,c}$ is the activity of TF motif $m$ in cell $c$; $\epsilon$ is Gaussian noise. The L2 penalty for ridge regression was determined automatically using an 80/20 cross-validation scheme. Z scores of motifs greater than 0.7 were kept as statistically significant motifs. Details can be found in Supplementary Note section Materials.

## Joint H3K4me1 and H3K9me3 analysis by double incubation

We assume that counts from double-incubated cells (H3K4me1 + H3K9me3) were generated by drawing $N$ reads from a mixture of two multinomials, one from a cell type $c$ from H3K4me1 (parametrized by relative frequencies $\vec{p}_c$) and one from a lineage $l$ from H3K9me3 (parametrized by relative frequencies $\vec{q}_l$):

$$\vec{y}|c,l,w \sim \text{Multinomial}\left(N, w\vec{p}_c + (1-w)\vec{q}_l\right),$$

where $w$ is the fraction of H3K4me1 that was mixed with H3K9me3. We used this model to calculate the likelihood that a double-incubated cell was generated by a specific pair of cell type and lineage combination. Details can be found in Supplementary Note section Materials.

## Imputing Sca1-cKit-Lin marker levels

Some cells had only two of the three marker levels (Sca1, cKit or Lin), and we imputed the missing third marker by averaging the top ten nearest neighbors in the cell that contains the missing marker levels. Details can be found in Supplementary Note section Materials.

## Reference-based cell typing using multinomials

We generated a ground truth reference dataset using FACS-defined labels, then used this reference to calculate the probability of each cell to be assigned to a cell type by assuming the counts from a cell were generated from a multinomial distribution parametrized by a cell type-specific vector of genomic locus probabilities. Details can be found in Supplementary Note section Materials.

## Inferring pseudotime across different differentiation trajectories

We manually selected two PCs for each cell type trajectory, selecting components that show large variation from progenitors (HSCs, LT, ST and MPPs), committed progenitors (for example, CMPs and MEPs), to mature cell types (for example, neutrophils, DCs, basophils, monocytes, pDCs, NK cells and B cells) of interest. Details can be found in Supplementary Note section Materials.

## Chromatin velocity in each histone modification

After defining a pseudotime for each differentiation trajectory, we fit a trajectory-specific cubic spline of the sortChIC signal along

pseudotime for each genomic region. We then calculate the derivative using the spline fits to predict the sortChIC signal of each cell at pseudotime $t$ to a future pseudotime $t + 0.01$. Details can be found in Supplementary Note section Materials[78].

## Reporting summary

Further information on research design is available in the Nature Portfolio Reporting Summary linked to this article.

## Data availability

Raw and processed data of this study are submitted to Gene Expression Omnibus (GEO) and available under accession number GSE164779. Public data used in this study can be found under K562 bulk ChIP data (H3K4me1, ENCSR000EWC; H3K4me3, ENCSR000EWA; H3K9me3, ENCSR000APE; H3K27me3, ENCSR000EWB), similar assays (GSE163532, GSE117309, GSE105012, GSE139857, GSE124557, GSE139857), scRNA-seq of mouse bone marrow (GSE113495) and TF motif database (http://swissregulon.unibas.ch/sr/downloads).

## Code availability

All processed and downstream scripts are available at https://github.com/jakeyeung/sortchicAllScripts (https://doi.org/10.5281/zenodo.7244251). Example vignettes to load and visualize the data are available at https://github.com/jakeyeung/sortChICAnalysis (https://doi.org/10.5281/zenodo.7108780). Downstream functions and standalone scripts to run latent Dirichlet allocation and infer TF activities are available at https://github.com/jakeyeung/scchic-functions (https://doi.org/10.5281/zenodo.7244208). The multinomial-based cell typing method AnnotateCelltypes is available as an R package at https://github.com/jakeyeung/AnnotateCelltypes (https://doi.org/10.5281/zenodo.7108451).

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

## Acknowledgements

We thank A. Giladi for sharing mRNA abundance tables of cell types together with J. van den Berg for critical reading of the manuscript. We thank M. Bartosovic for sharing method comparison data. pK19pA-MN was a gift from Ulrich Laemmli (Addgene plasmid 86973, http://n2t.net/addgene:86973; RRID:Addgene_86973). Figure 8 is adopted from Hematopoiesis (human) diagram by A. Rad and M. Häggström under CC-BY-SA 3.0 license. This work was supported by European Research Council Advanced under grant ERC-AdG 742225-IntScOmics and Nederlandse Organisatie voor Wetenschappelijk Onderzoek (NWO) TOP award NWO-CW 714.016.001. The SNF (P2BSP3-174991), HFSP (LT000209/2018-L) and Marie Skłodowska-Curie Actions (798573) supported P.Z. The SNF (P2ELP3_184488) and HFSP (LT000097/2019-L) supported J.Y. and the EMBO LTF (ALTF 1197–2019) supported V.B. This work is part of the Oncode Institute, which is partly financed by the Dutch Cancer Society. The funders had no role in study design, data collection and analysis, decision to publish or preparation of the manuscript.

## Author contributions

P.Z., J.Y. and A.v.O. designed the project; P.Z. and M.F. developed experimental methods; P.Z., H.V.G. and R.v.d.L. performed experiments; J.Y. developed and applied the statistical methods. B.A.d.B., M.F., and J.Y. wrote the sortChIC demultiplexing and preprocessing pipeline. J.Y., A.v.O, P.Z. and V.B. analyzed the data. P.Z., J.Y. and A.v.O, wrote the manuscript.

## Competing interests

The authors declare no competing interests.

## Additional information

**Extended data** is available for this paper at https://doi.org/10.1038/s41588-022-01260-3.

**Correspondence and requests for materials** should be addressed to Alexander van Oudenaarden.

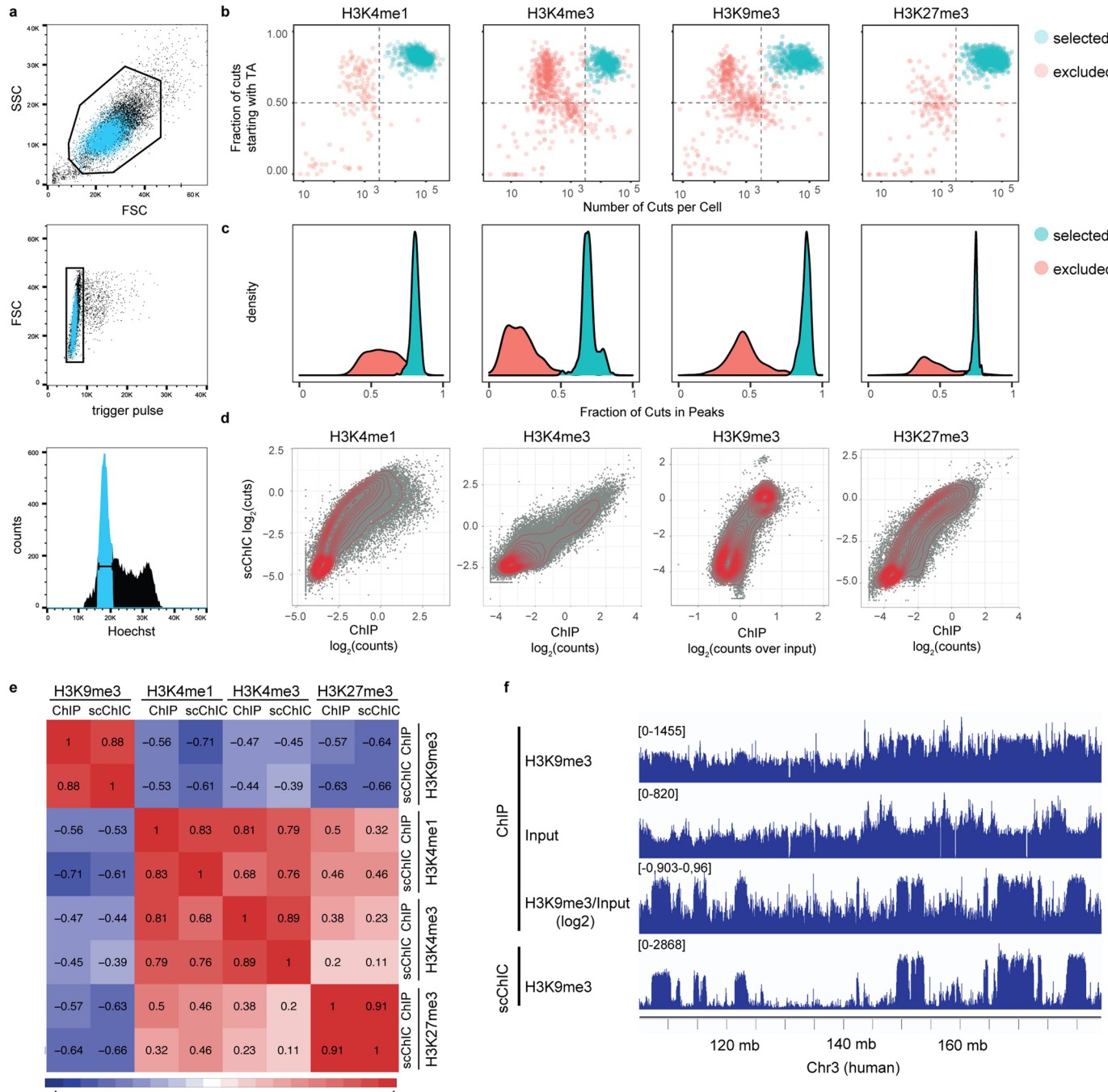

**Extended Data Fig. 1 | sortChIC generates high-resolution maps of histone modifications in single cells.** (**a**) FACS plots for sorting individual K562 cells in G1 phase. (**b**) Fraction of cuts starting with TA (reflecting the preference of MNase to cut in an AT context) versus number of cuts mapped to the K562 genome. Cells below horizontal dotted lines and left of vertical lines are excluded from the analysis. (**c**) Distribution of fraction of cuts mapped to locations within peaks across cells. (**d**) Correlation between pseudobulk sortChIC and bulk ChIP signal using 50 kilobase (kb) bins for H3K4me1, H3K4me3, H3K27me3, and H3K9me3. (**e**) Pearson correlation between pseudobulk sortChIC and bulk ChIP signal using 50 kb bins across the four histone marks. (**f**) Three tracks of H3K9me3 ChIP-seq bulk data, one for H3K9me3 without normalization (H3K9me3), one for the input (Input), and one where H3K9me3 is normalized to the input (H3K9me3/input). Fourth track is H3K9me3 sortChIC pseudobulk, showing that H3K9me3 ChIP-seq requires normalizing by input to resemble sortChIC.

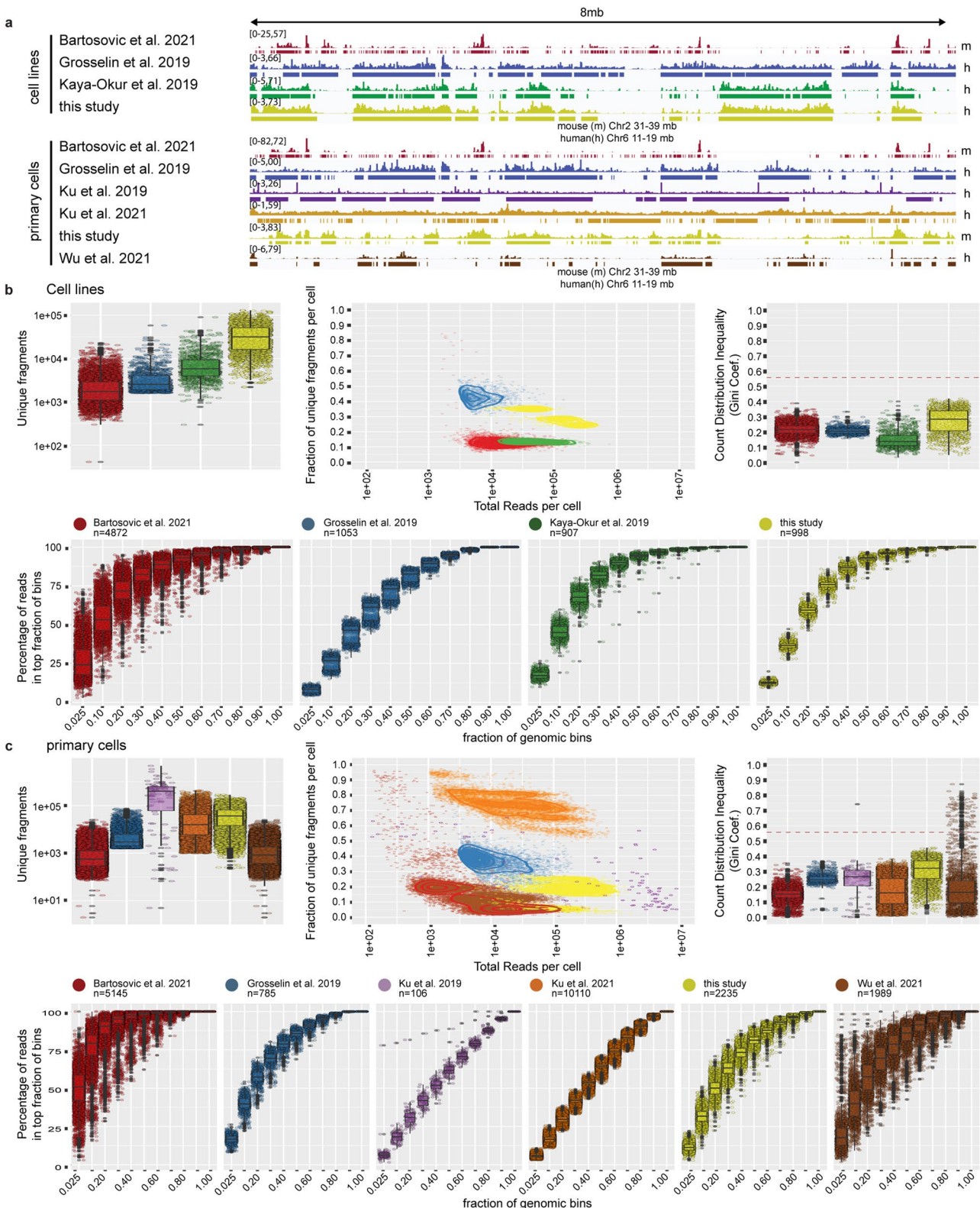

**Extended Data Fig. 2 | Comparison with existing single cell chromatin profiling methods.** (a) genomic tracks of pseudobulk data of either cell lines (top) or primary cells (bottom). Tracks labeled on the right with h show 8 mb of the human chromosome 6 (Chr6:11–19 mb). Tracks labeled with m show 8 mb of the mouse chromosome 2 (Chr2: 31–39 mb). Lines underneath each track indicate peak calling results. (**b, c**) Comparison across studies of unique fragments per cell, fraction of unique reads vs mapped reads, Gini coefficient, and cumulative distribution of signal over the genome. The spread of data points per genomic fraction reflects agreement between single cells, the elbow-point indicates the fraction of the genome covered by the histone mark. (**b**) cell lines. (**c**) primary cells. Boxplots show 25th percentile, median and 75th percentile, with the whiskers spanning 97% of the data. Red line in Gini plots indicates coefficient determined from public bulk chip sample (ENCSR000EWB).

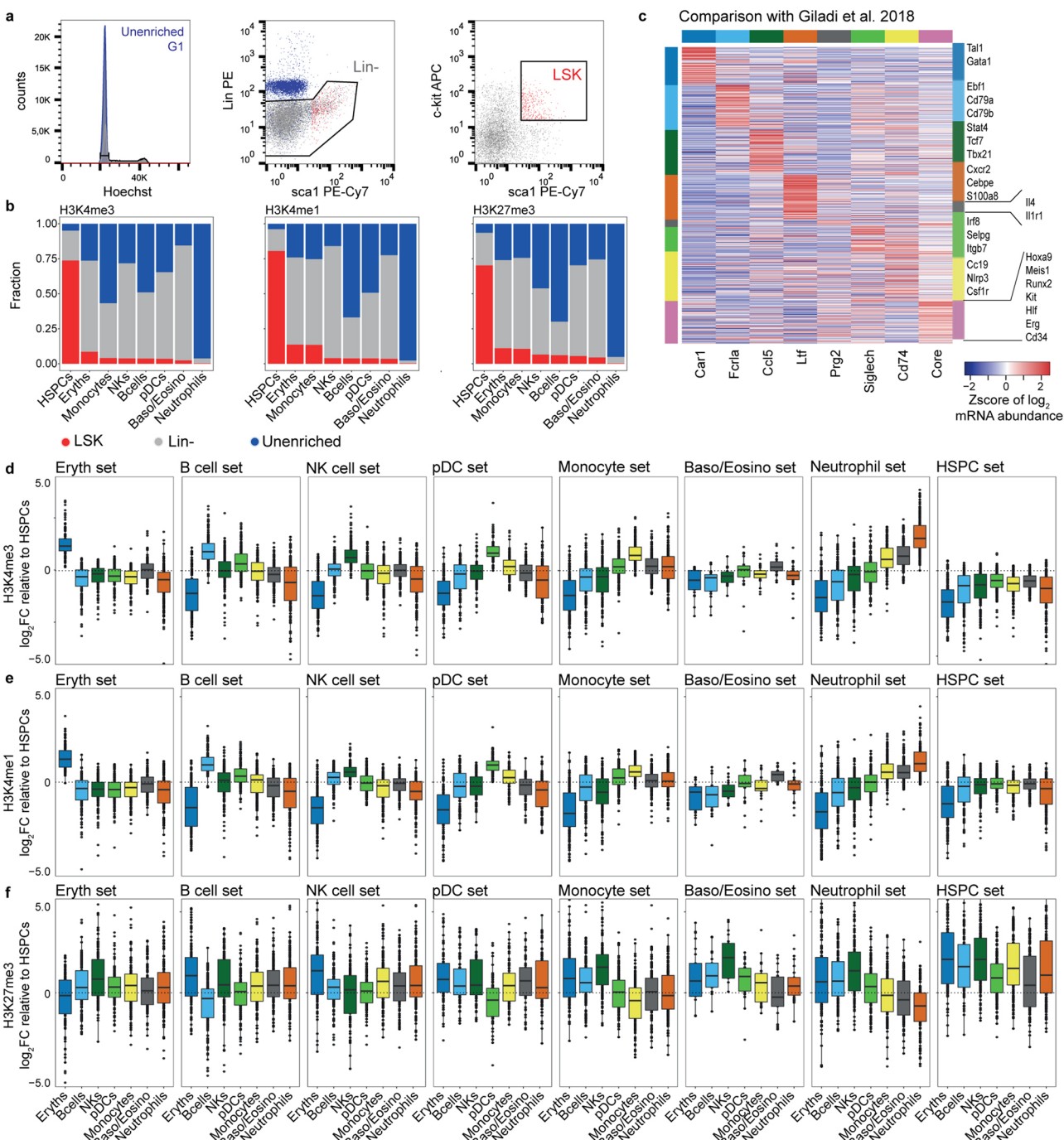

**Extended Data Fig. 3 | H3K4me1 and H3K4me3 in HSPCs prime for different blood cell fates, while H3K27me3 in differentiated cell types silences genes of alternative cell fates.** (**a**) FACS plot for sorting G1 cells of whole bone marrow (unenriched), lineage negative (Lin−), and Lin−,Sca1+, cKit+ (LSK) populations. (**b**) Fraction of cells in each cell type labeled by the sorted population: whole bone marrow (unenriched), lineage negative (Lin−), and Lin−Sca1+cKit+ (LSK). (**c**) Cell type-specific mRNA abundances for genes associated with regions in Fig. 2E using pseudobulk analysis of the Giladi et al. 2018 dataset (Methods). (**d**) H3K4me3 fold changes of different cell types relative to HSPCs at cell type-

specific regions. Each panel corresponds to a set of cell type-specific regions defined by the rows of one color in the heatmap of Fig. 2e. Regions are defined by +/− 5 kilobase windows centered at transcription start sites of cell type-specific genes. (**e**) Same as (d) but for H3K4me1. (**f**) Same as (d) but for H3K27me3. Boxplots show 25th percentile, median and 75th percentile, with the whiskers spanning 97% of the data. For DCs and Baso/Eosino sets, each boxplot contains n = 91 and n = 25 regions, respectively. For all other sets, each boxplot contains n = 150 regions.

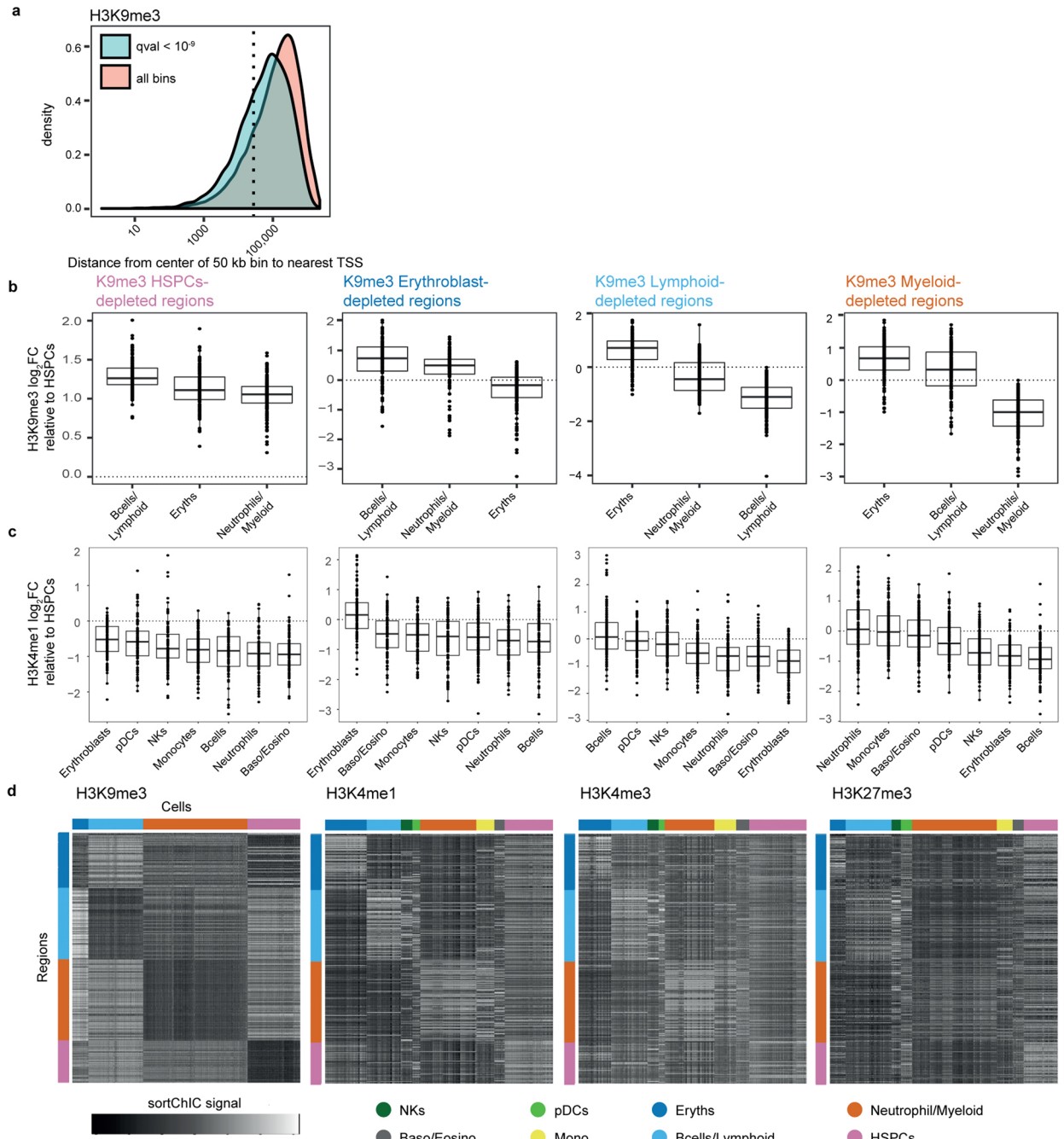

**Extended Data Fig. 4 | Lineage-specific loss of H3K9me3 correlates with cell type-specific increase in H3K4me1. (a)** Statistically significant 50 kb regions (adjusted p-value < 109, deviance goodness-of-fit test) identified for H3K9me3, showing distribution of distances from center of 50 kb region to nearest TSS of a gene. All bins are identified as 50 kb regions that have pseudobulk (counts summed across all cells) signal above background levels (Methods). Dotted line represents 25 kb, meaning the bin would overlap with a TSS. **(b)** Fold change in H3K9me3 relative to HSPCs for four sets of 150 regions: regions depleted in erythroblasts, lymphoid, myeloid, or HSPCs. Each region is 50 kb wide. Each boxplot contains n = 150 regions. **(c)** The same four sets of regions but showing fold change in H3K4me1, showing upregulation of H3K4me1 specifically in cell types that are depleted in H3K9me3. Each boxplot contains n = 150 regions. Boxplots show 25th percentile, median and 75th percentile, with the whiskers spanning 97% of the data. **(d)** Heatmap of the four regions in single cells across the four marks. Rows are regions, color coded as in top of (b). Columns are cells, color coded as shown below.

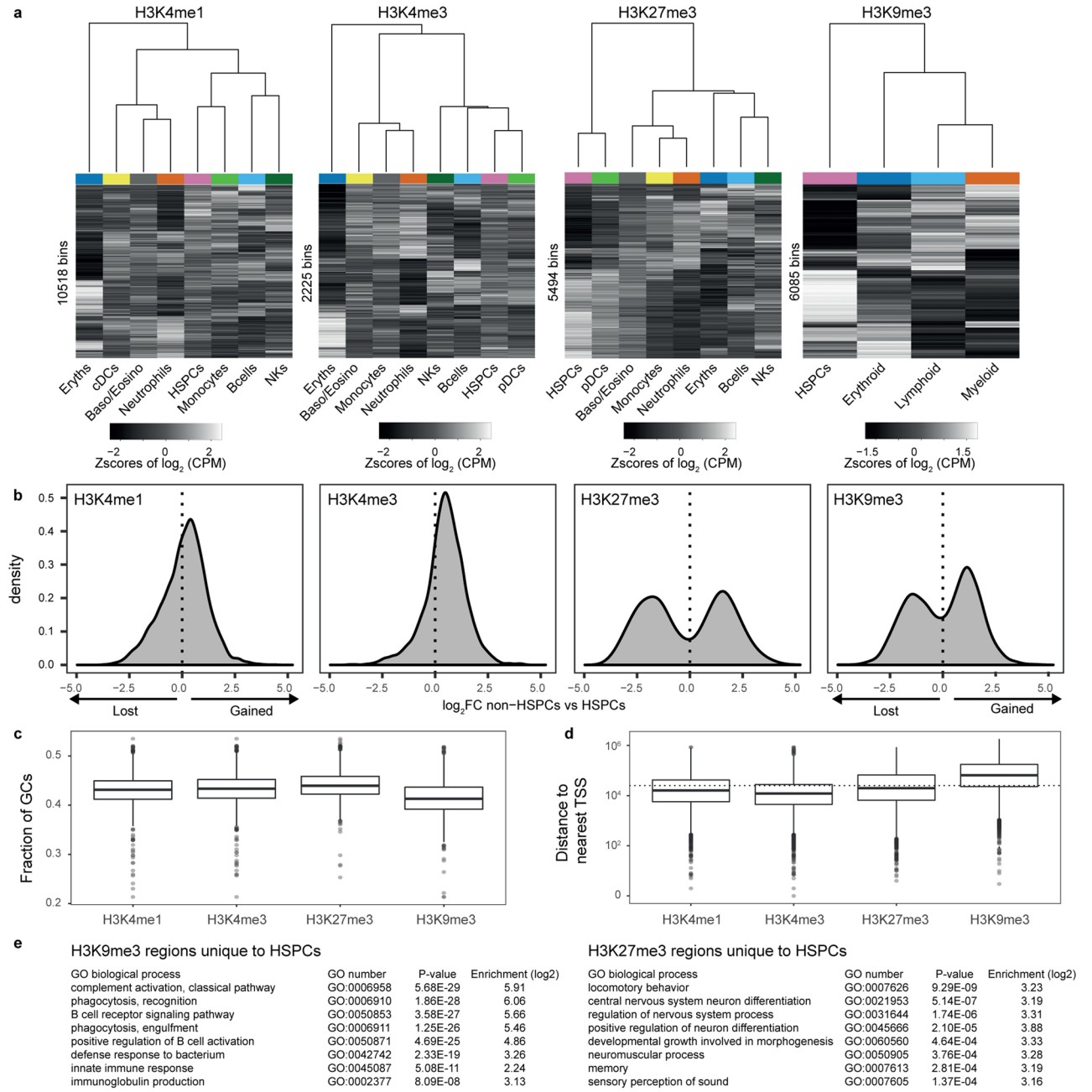

**Extended Data Fig. 5 | Features of active and repressive chromatin dynamics during hematopoiesis.** (**a**) Heatmap of log2 counts per million (CPM) of 50 kilobase bins across pseudobulks. Changing bins that are statistically significant are shown (deviance goodness-of-fit test from Poisson regression, Methods). The rows and columns are ordered by complete-linkage clustering. Above each heatmap is a dendrogram from clustering the columns, showing the relationship between cell types. (**b**) Distribution of log2 fold changes (FC) at statistically significant changing bins (null model: a bin has constant signal across all cell types, full model: a bin has signal that depends on cell type, deviance goodness-of-fit test) between pseudobulk of non-HSPCs versus HSPCs. Bimodal distribution highlights differences originate mainly between HSPCs and non-HSPCs. (**c**) GC content of dynamic 50 kb bins for the four histone marks. Number of dynamic bins depends on the mark. H3K4me1: n = 10518 bins; H3K4me3: n = 2225 bins; H3K27me3: n = 5494 bins; H3K9me3: n = 6085 bins. (**d**) Distance to nearest TSS measured from the center of each dynamic 50 kb bin. Dotted horizontal line represents 25 kb, meaning the bin would overlap with a TSS. Boxplots show 25th percentile, median and 75th percentile, with the whiskers spanning 97% of the data. (**e**) Gene ontology (GO) terms of HSPC-specific H3K9me3 (top) and H3K27me3 (bottom) regions. P-value and enrichment from Fisher's exact test.

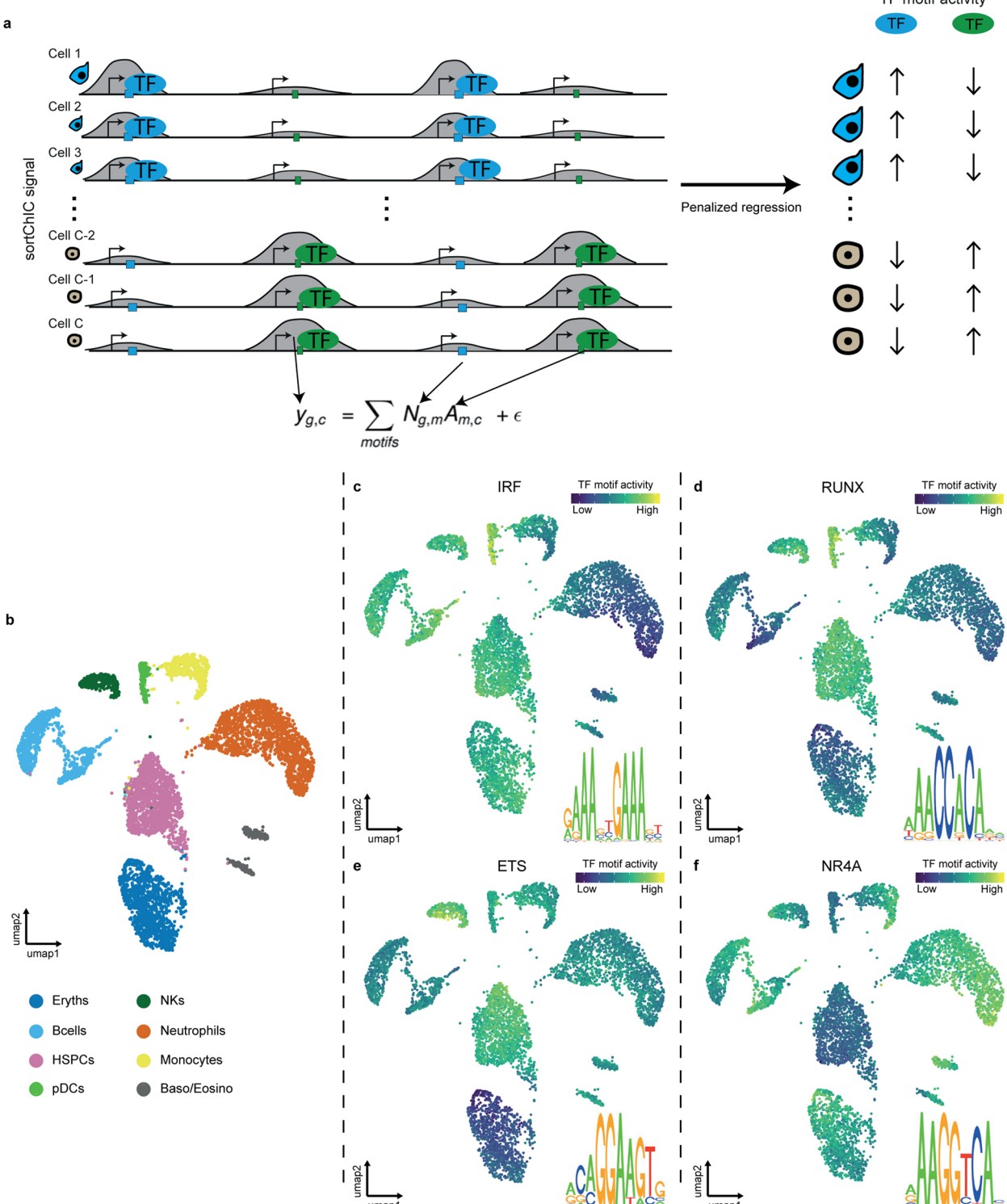

**Extended Data Fig. 6 | Penalized regression model reveals transcription factor motifs underlying cell type-specific chromatin dynamics. (a)** Schematic of the transcription factor (TF) activity model. The penalized regression model takes the imputed sortChIC signal in a peak as the response variable and the TF binding motifs predicted under each peak as the explanatory variable (Method). The penalized multivariate regression infers the TF motif activity driving cell type-specific sortChIC signal. (**b**) UMAP of H3K4me1 chromatin states in single cells, colored by cell type. (**c–f**) UMAP where each cell is colored by the TF activity inferred from the model. Four cell type-specific TF motifs are shown.

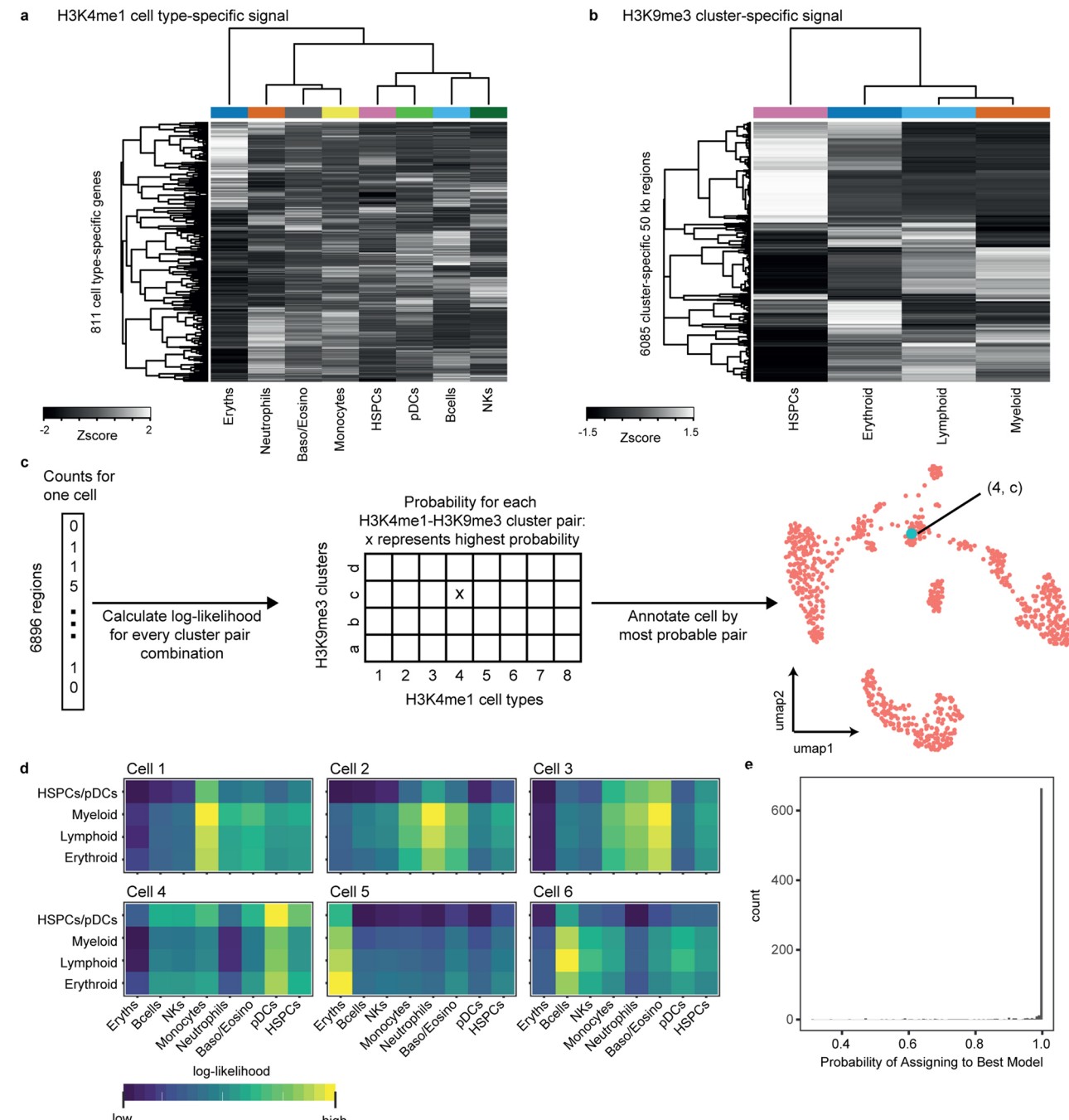

**Extended Data Fig. 7 | Single-incubated data from H3K4me1 and H3K9me3 builds a model for inferring cluster-pairs in double-incubated data. (a)** Heatmap of H3K4me1 signal across clusters for 811 cell type-specific regions (Methods). These regions come from cell type-specific genes used in Fig. 2e. (**b**) Heatmap of H3K9me3 signal across clusters for 6085 cluster-specific regions (50 kb genomic window). These regions come from the statistically significant dynamic regions of H3K9me3 defined in Extended Data Fig. 4a. (**c**) Schematic of how a cluster-pair is inferred from each double-incubated cell. Each double-incubated cell has a vector of counts across 6896 regions (811 regions come from H3K4me1, while 6085 come from H3K9me3). We calculate the log-likelihood (Methods) of the observed double-incubated cell counts for each cluster-pair (32 cluster-pairs from 8 clusters in H3K4me1 and 4 clusters in H3K9me3). From the 32 log-likelihood estimates, we assign the cell to the cluster-pair with the highest probability. (**d**) Examples of the 32 log-likelihood estimates from eight representative cells, shown as a 4-by-8 heatmap. Each of the four rows is a cluster from H3K9me3; each of the eight columns is a cluster from H3K4me1. (**e**) Histogram of the highest assignment probability per cell.

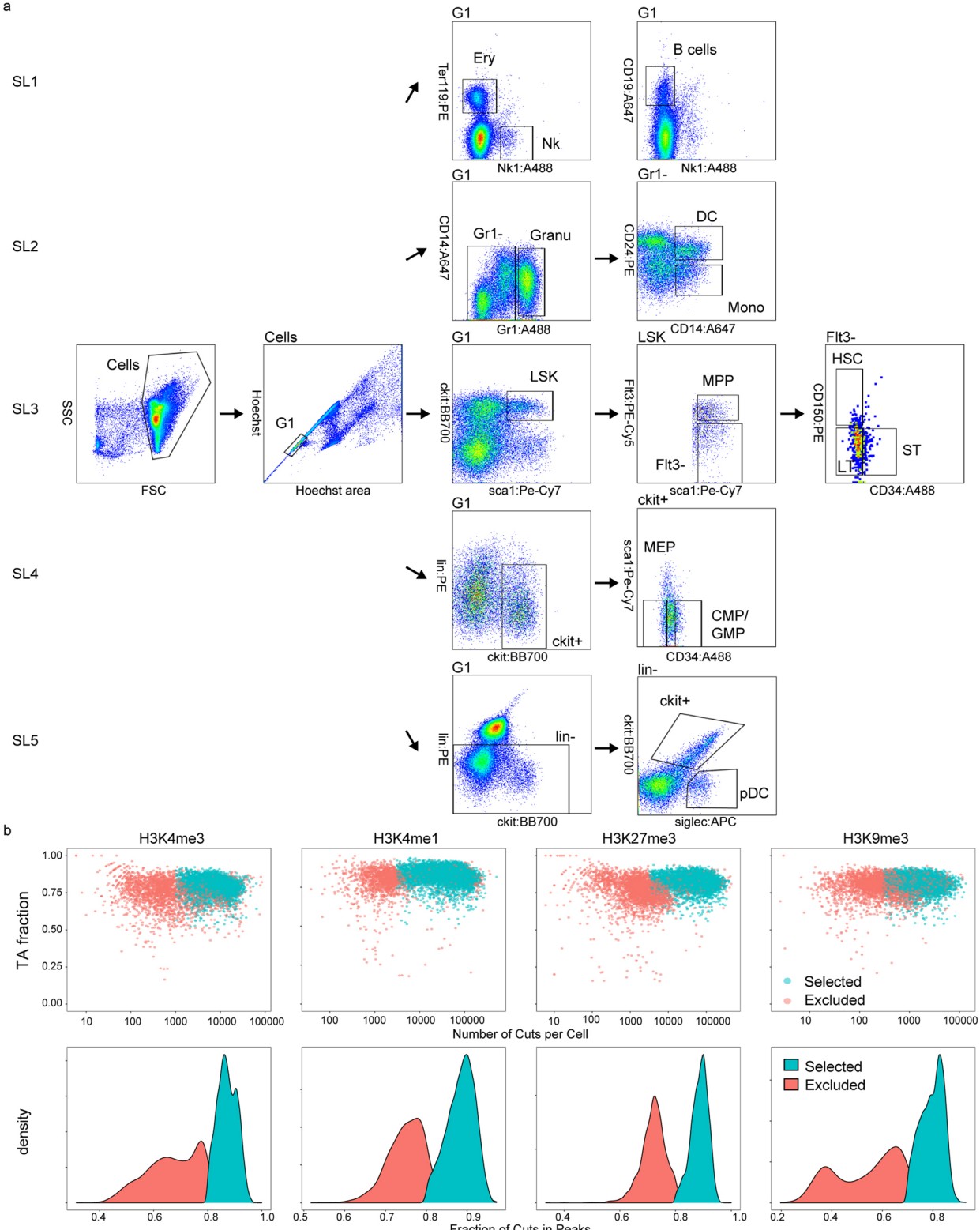

**Extended Data Fig. 8 | FACS gating and quality control for sortChIC design spanning HSCs, progenitors, and mature cell types.** (**a**) FACS gating plots for the five sorting strategies in the expanded sortChIC experiment. (**b**) Scatter plots of number of unique cuts against fraction of reads starting with MN specific TA per cell (top) and density plot of fraction of cuts in peaks (bottom) of the sortChIC data across the four marks, split by included or excluded by quality control cutoffs.

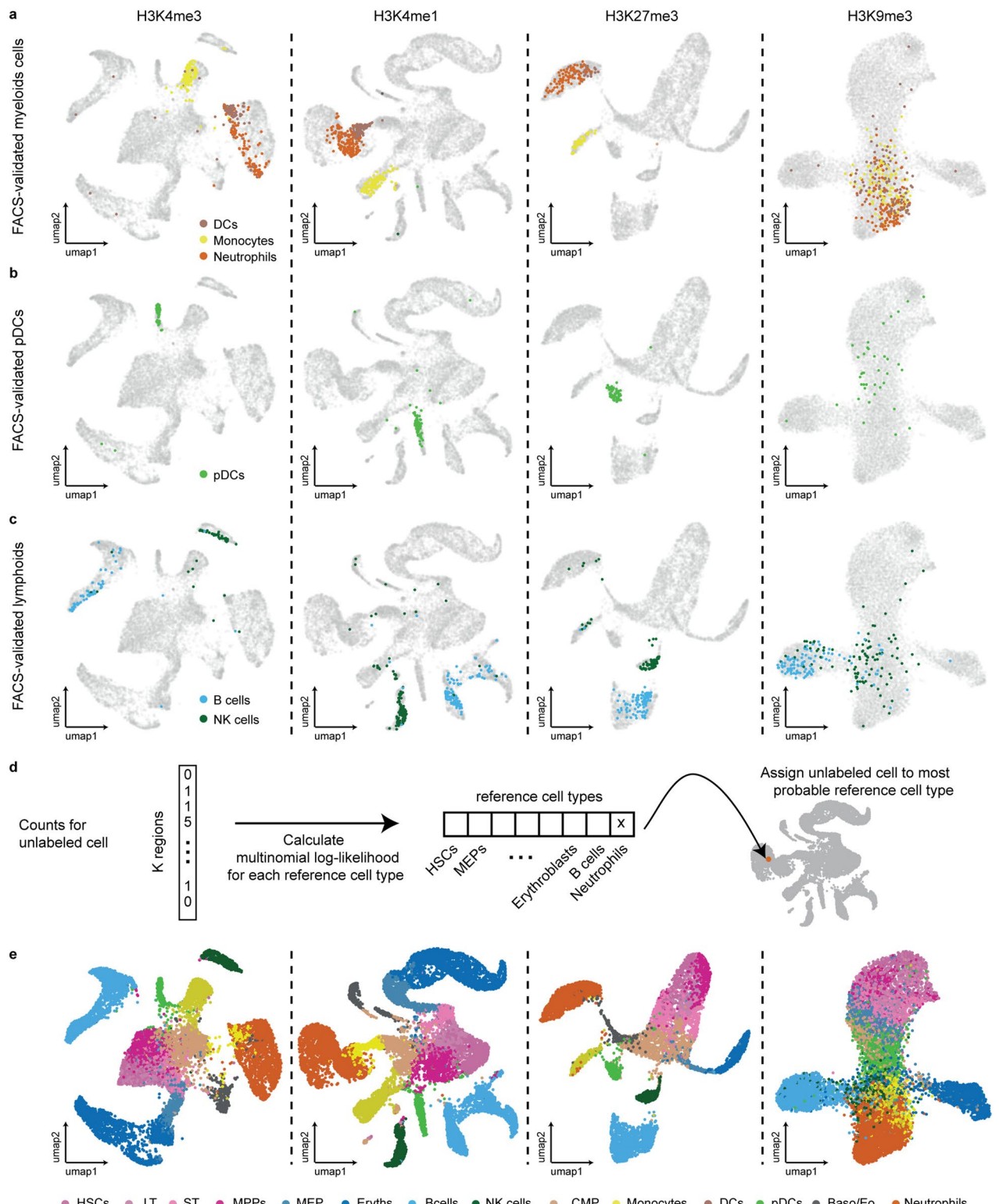

**Extended Data Fig. 9 | FACS gating based UMAP annotation.** (**a**–**c**) UMAP of the combined sortChIC data with the position of a selection of FACS sorted cells highlighted. neutrophils, monocytes, and DCs are labeled in (a) pDCs in (b) and B and NK cells in (c). (**d**) Schematic illustrating how reference cell types are used to systematically assign the rest of the cells without FACS-defined cell type label. (**e**) Final UMAP of the bone marrow data across four histone marks after assigning cell types to one of the 14 cell types in the reference.

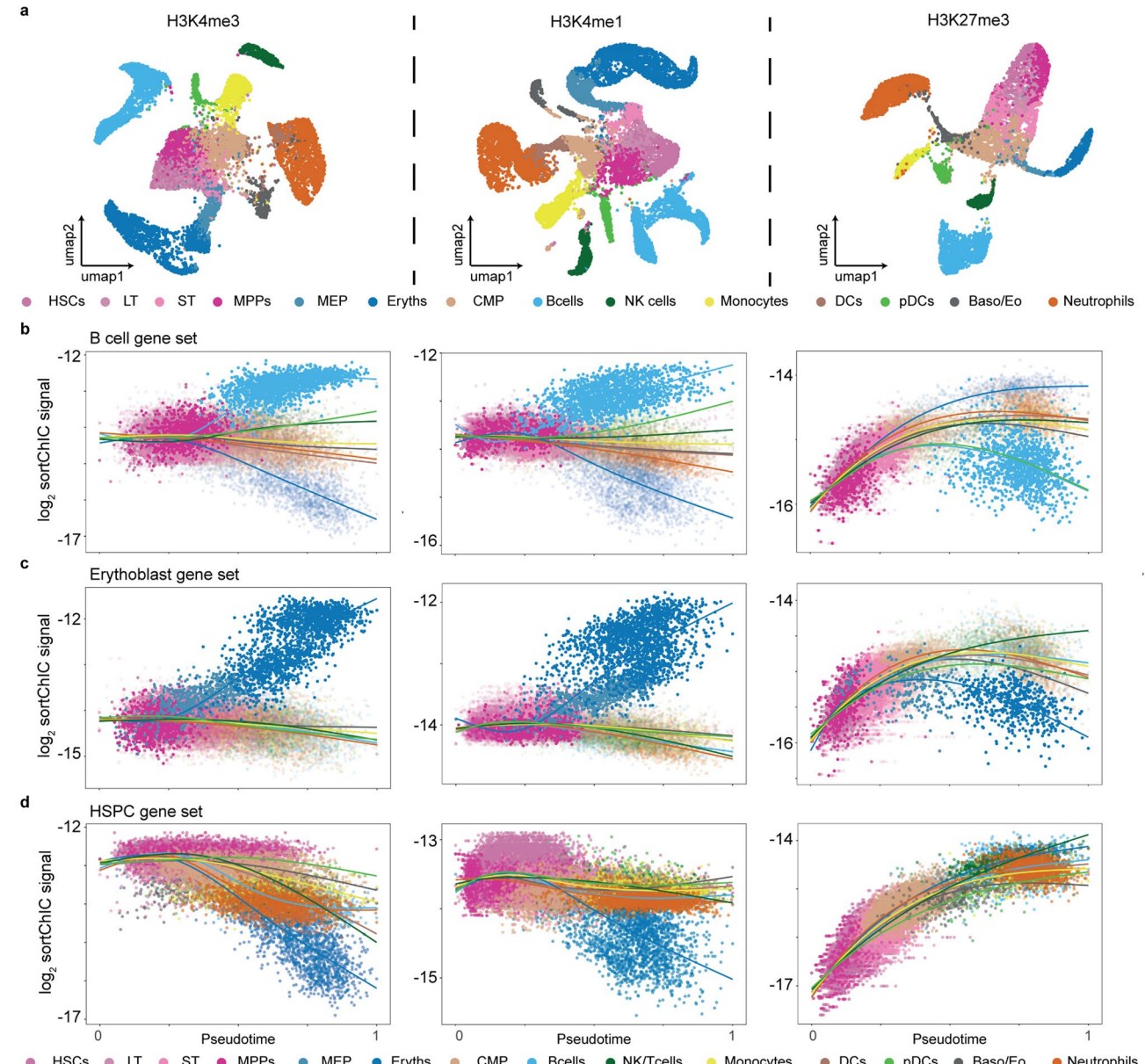

**Extended Data Fig. 10 | Pseudotime analysis from HSCs to mature cell types at TSS of cell-type specific genes.** (**a**) UMAP for H3K4me3 (n = 12085), H3K4me1 (n = 10952) and H3K27me3 (n = 7984). (**b**) H3K4me3, H3K4me1, and H3K27me3 mean sortChIC signal across B cell-specific marker genes (the same 150 marker genes defined from heatmap from Fig. 2e). (**c**) Same as (b) but for erythroblast marker genes. (**d**) Same as (b) but for HSPC marker genes.

# Reporting Summary

## Statistics

For all statistical analyses, confirm that the following items are present in the figure legend, table legend, main text, or Methods section.

| n/a | Confirmed | |
|-----|-----------|---|
| ☐ | ☒ | The exact sample size ($n$) for each experimental group/condition, given as a discrete number and unit of measurement |
| ☐ | ☒ | A statement on whether measurements were taken from distinct samples or whether the same sample was measured repeatedly |
| ☐ | ☒ | The statistical test(s) used AND whether they are one- or two-sided *Only common tests should be described solely by name; describe more complex techniques in the Methods section.* |
| ☒ | ☐ | A description of all covariates tested |
| ☐ | ☒ | A description of any assumptions or corrections, such as tests of normality and adjustment for multiple comparisons |
| ☐ | ☒ | A full description of the statistical parameters including central tendency (e.g. means) or other basic estimates (e.g. regression coefficient) AND variation (e.g. standard deviation) or associated estimates of uncertainty (e.g. confidence intervals) |
| ☐ | ☒ | For null hypothesis testing, the test statistic (e.g. $F$, $t$, $r$) with confidence intervals, effect sizes, degrees of freedom and $P$ value noted *Give P values as exact values whenever suitable.* |
| ☐ | ☒ | For Bayesian analysis, information on the choice of priors and Markov chain Monte Carlo settings |
| ☒ | ☐ | For hierarchical and complex designs, identification of the appropriate level for tests and full reporting of outcomes |
| ☐ | ☒ | Estimates of effect sizes (e.g. Cohen's $d$, Pearson's $r$), indicating how they were calculated |

*Our web collection on statistics for biologists contains articles on many of the points above.*

## Software and code

Policy information about availability of computer code

| Data collection | single cell FACS data was collected using BD FACS software (version 1.2.0.124) |
|-----------------|-------------------------------------------------------------------------------|
| Data analysis | Github repo of processing and analysis scripts: https://github.com/jakeyeung/sortChIC<br><br>bwa Heng et al., 2010 Bioinformatics Version: 0.7.17-r1188<br>topicmodels Hornik & Grun 2011 Journal of Statatistical Software Version: 0.2-12<br>Glmpca Townes et al. 2019 Genome Biology Version: 0.2.0<br>motevo Arnold et al. 2012 Bioinformatics Version: 1.11<br>hiddenDomains Starmer and Magnuson 2016 BMC Bioinformatics Version: 3.1<br>SingleCellMultiOmics https://github.com/BuysDB/SingleCellMultiOmics/wiki |

For manuscripts utilizing custom algorithms or software that are central to the research but not yet described in published literature, software must be made available to editors and reviewers. We strongly encourage code deposition in a community repository (e.g. GitHub). See the Nature Portfolio guidelines for submitting code & software for further information.

## Data

Policy information about availability of data

All manuscripts must include a data availability statement. This statement should provide the following information, where applicable:

- Accession codes, unique identifiers, or web links for publicly available datasets
- A description of any restrictions on data availability
- For clinical datasets or third party data, please ensure that the statement adheres to our policy

All data produced in this study are deposited onto GEO under accession GSE164779. Token "srmnyauiphmflmx"
Data for K562 specificity comparison:
H3K4me1, Peggy Farnham, ENCSR000EWC, pAb-037-050 (Diagenode), H3K4me3, Peggy Farnheim, ENCSR000EWA, 9751S (Cell Signaling)
H3K9me3, Bradley Bernstein, ENCSR000APE, ab8898 (Abcam), H3K27me3, Peggy Farnheim, ENCSR000EWB, 9733S (Cell Signaling)
Data for comparison with similar assays was downloaded from GEO: Bartosovic et. al. (GSE163532), Grosselin et. al. (GSE117309), Ku et. al. 2019 (GSE105012), Wu. et. al. (GSE139857), Kaya-Okur et. al. (GSE124557), and Ku et. al. 2021 (GSE139857)
Transcription factor binding motives were used from the mm10 Swiss Regulon database of 680 motifs (http://swissregulon.unibas.ch/sr/downloads)
bone marrow scRNAseq data was used from Giladi et al 2018 (GSE113495)

# Field-specific reporting

Please select the one below that is the best fit for your research. If you are not sure, read the appropriate sections before making your selection.

☒ Life sciences          ☐ Behavioural & social sciences          ☐ Ecological, evolutionary & environmental sciences

For a reference copy of the document with all sections, see nature.com/documents/nr-reporting-summary-flat.pdf

# Life sciences study design

All studies must disclose on these points even when the disclosure is negative.

| | |
|---|---|
| Sample size | Sample sizes (number of single cells) were chosen based on number of cells per cluster and occurrence of new clusters. They were determined to be sufficient when each cell type specific cluster contained at least 20 cells, and no new cell clusters were detected after doubling sample size. |
| Data exclusions | Single cells were selected based on minimal unique fragments per cell (500 for H3K4me1 and H3K4me3, 1000 for H3K9me3 and H3K27me3), at least 50% of the reads per cell had to start with the MN specific AT bias and intra-chromsomal variance of maximally 2 fold above average. More details can be found in Supplementary fig1 and the methods section. |
| Replication | Results were reproducible over 2 independent biological replica (2 mice) and hundreds of linked biological replicas (thousands of cells) |
| Randomization | No randomization was performed as there were no treatments/experiments were performed on the animals before cell isolation |
| Blinding | No blinding was performed as only one animal strain was used and results from all experiments were analysed with the identical computational pipeline, ensuring no influence of the experimentalist on observed differences. |

# Reporting for specific materials, systems and methods

We require information from authors about some types of materials, experimental systems and methods used in many studies. Here, indicate whether each material, system or method listed is relevant to your study. If you are not sure if a list item applies to your research, read the appropriate section before selecting a response.

## Materials & experimental systems

| n/a | Involved in the study |
|---|---|
| ☐ | ☒ Antibodies |
| ☐ | ☒ Eukaryotic cell lines |
| ☒ | ☐ Palaeontology and archaeology |
| ☐ | ☒ Animals and other organisms |
| ☒ | ☐ Human research participants |
| ☒ | ☐ Clinical data |
| ☒ | ☐ Dual use research of concern |

## Methods

| n/a | Involved in the study |
|---|---|
| ☐ | ☒ ChIP-seq |
| ☐ | ☒ Flow cytometry |
| ☒ | ☐ MRI-based neuroimaging |

## Antibodies

| | |
|---|---|
| Antibodies used | For ChIP (sortChIC): |

| Antibodies used | H3K4mel, ab8895 (Abeam), Lot: GR3206285-1;<br>H3K4me3, 07-473 (Merck), Lot: 3093304;<br>H3K4me3, MA5-11199 (Thermo Fisher), clone: G.532.8, monoclonal;<br>H3K9me3, ab8898 (Abeam), Lot: GR3217826-1;<br>H3K9me3, MA5-33395 (Thermo Fisher), clone: RM389, monoclonal;<br>H3K27me3, 9733S (NEB), clone: C36B11, monoclonal<br>For Flow cytometry:<br>C-kit-APC, 105811 (Biolegend), clone: 2B8, monoclonal;<br>Sca1-PeCy7, 108113 (Biolegend), clone: D7, monoclonal;<br>NK1-Alexa488, 108717 (Biolegend), clone: PK136, monoclonal;<br>Ter119-PE, 116207 (Biolegend), clone: Ter-119, monoclonal;<br>CD19-Alexa647, 557684 (BD), clone: 1D3, monoclonal;<br>CD3-APC-Cy7, 100221 (Biolegend), clone: 17A2, monoclonal;<br>CD11b-APC-Cy7, 561039 (BD), clone: M1/70, monoclonal;<br>CD14-Alexa647, 565743 (BD), clone: rmC5-3, monoclonal;<br>CD24-PE, 101807 (Biolegend), clone: M1/69, monoclonal;<br>Grl-Alexa488, 53-5931-80 (Thermo Fisher), clone: 53-5931-8, monoclonal;<br>C-kit-BB700, 566414 (BD), clone: 2B8, monoclonal;<br>Flt3-PE-Cy5, 15-1351-82 (Thermo Fisher), clone: A2F10, monoclonal;<br>CD150-PE, 562651 (BD), clone: Q38-480, monoclonal;<br>CD34-Alexa488, 53-0341-82 (Thermo Fisher), clone: RAM34, monoclonal;<br>FCgammaR-APC, 17-0161-81 (Thermo Fisher), clone: 93, monoclonal;<br>Siglec-APC, 17-0333-80 (Thermo Fisher), clone: eBio440c, monoclonal;<br>IL7R-Alexa488, 53-1271-82 (Thermo Fisher), clone: A7R34, monoclonal; |
|---|---|
| Validation | For the ChIP experiments<br>ab8895, 07-473, ab8898 and C36B11 where validated by provider by ChIP-seq in mouse<br>MA5-11199 and MA5-33395 where validated by provider for specificity by peptide array and Eliza respectively<br>all antibodies where further validated in this study by correlation to public ChIP data<br>data for ab8895, MA5-11199, MA5-33395 and 9733S are shown in extended Fig 1<br>All antibodies used for FACS analysis where validated by provider for FACS application in mouse |

# Eukaryotic cell lines

Policy information about cell lines

| Cell line source(s) | K562, CCL-243 (ATCC) |
|---|---|
| Authentication | authentication by the supplier by Karyotyping and antigen expression |
| Mycoplasma contamination | Cell lines were tested every 2 months and never tested positive |
| Commonly misidentified lines (See ICLAC register) | no commonly misidentified cell lines were used in this study |

# Animals and other organisms

Policy information about studies involving animals; ARRIVE guidelines recommended for reporting animal research

| Laboratory animals | C57BL/6 mice, male, 3-months-old<br>All mouse studies were conducted in accordance with protocols approved by the ethics committee of the Hubrecht Institute in Utrecht. Mice were housed in a normal condition with 12:12h light: dark cycle in a temperature-controlled room with food and water ad libitum. |
|---|---|
| Wild animals | No wild animals were used in this study |
| Field-collected samples | no Field collected samples were used in this study |
| Ethics oversight | Dier Experimenten Commissie of the Royal Netherlands Academy of Arts and Sciences |

Note that full information on the approval of the study protocol must also be provided in the manuscript.

# ChIP-seq

## Data deposition

☒ Confirm that both raw and final processed data have been deposited in a public database such as GEO.

☐ Confirm that you have deposited or provided access to graph files (e.g. BED files) for the called peaks.

| Data access links<br>*May remain private before publication.* | https://www.ncbi.nlm.nih.gov/geo/query/acc.cgi?acc=GSE164779 |
|---|---|

| Files in database submission | tagged bam files split by cell-type, batch corrected metadata and 50kb count tables are provided for each experiment, including K562 data for H3K4me1, H3K4me3, H3K9me3 and H3K27me3, as well as BM data merged from 2 independent biological replicas including H3K4me1, H3K4me3, H3K9me3, H3K27me3 and H3K4me1+H3K9me3 double incubation |
| --- | --- |
| Genome browser session (e.g. UCSC) | NA |

## Methodology

| Replicates | Cell line experiments were performed in 3 independent biological replica (N=3) with hundreds of single cell observation (n=276) per replica. Mouse experiments were performed as 2 independent biological replica (N=2) with thousands of single cell observations (n=~3000) per replica. |
| --- | --- |
| Sequencing depth | Samples were sequenced to a depth of at least 1.5 oversequencing rate. Unique number of fragments per cell are provided in the manuscript. Sequencing was performed paired end 75 bp |
| Antibodies | H3K4me1, ab8895 (Abcam), Lot: GR3206285-1; H3K4me3, 07-473 (Merck), Lot: 3093304; H3K4me3, MA5-11199 (Thermo Fisher), monoclonal; H3K9me3, ab8898 (Abcam), Lot: GR3217826-1; H3K9me3, MA5-33395 (Thermo Fisher), monoclonal; H3K27me3, 9733S (NEB), monoclonal |
| Peak calling parameters | details on read mapping can be found in the methods section of the manuscript and https://github.com/BuysDB/SingleCellMultiOmics/wiki. For peak calling hiddenDomains was used with minimum peak length of 1000 bp |
| Data quality | NA |
| Software | bwa  Heng et al., 2010 Bioinformatics Version: 0.7.17-r1188<br>topicmodels Hornik & Grun 2011 Journal of Statatistical Software Version: 0.2-12<br>Glmpca Townes et al. 2019 Genome Biology Version: 0.2.0<br>motevo Arnold et al. 2012 Bioinformatics Version: 1.11<br>hiddenDomains Starmer and Magnuson 2016 BMC Bioinformatics Version: 3.1<br>SingleCellMultiOmics https://github.com/BuysDB/SingleCellMultiOmics/wiki |

# Flow Cytometry

## Plots

Confirm that:

☒ The axis labels state the marker and fluorochrome used (e.g. CD4-FITC).

☒ The axis scales are clearly visible. Include numbers along axes only for bottom left plot of group (a 'group' is an analysis of identical markers).

☒ All plots are contour plots with outliers or pseudocolor plots.

☒ A numerical value for number of cells or percentage (with statistics) is provided.

## Methodology

| Sample preparation | Where indicated  mouse bone marrow cells were stained live with antibody combinations (see methods section) for surface marker staining. Cell culture cells and mouse bone marrow cells were further washed in PBS and fixed in 70% ethanol 1h at -20C. Cells were incubated over night with histone mark specific antibodies at 4C. Before the sorting unbound antibody was washed away and the Pa-MN fusion protein targeting and Hoechst staining was performed for 1h at 4C. After 2 extra washes cell were sorted in 384 well plates. More details see methods section. |
| --- | --- |
| Instrument | BD Influx™ Cell Sorter |
| Software | For FACS, BD FACS software (version 1.2.0.124) was used. |
| Cell population abundance | purity in post-sorted samples is determined by single cell sequencing and shown in the manuscript |
| Gating strategy | Gating strategy is illustrated in the manuscript Extended Figure 1a, 2a and 8a. For Cell lines are gated for SSC, FSC and a G1 hoechst stain. For Bone marrow cells the gating strategy includes FSC, SSC and a G1 hoechst staining for all sorted cells, +linage marker negative for lin- cells, +cKit positive and sca positive for LSK cells. |

☒ Tick this box to confirm that a figure exemplifying the gating strategy is provided in the Supplementary Information.

