## [Peer Review File · Nature Genetics]

Peer Review Information

Manuscript Title: Hierarchical chromatin regulation during blood formation uncovered by single-cell sortChIC

Corresponding author name(s): Alexander van Oudenaarden

Reviewer Comments & Decisions:

Decision Letter, initial version:
--

Dear Professor van Oudenaarden,

Thank you for submitting your manuscript entitled "Hierarchical chromatin regulation during blood formation uncovered by single-cell sortChIC". We have given the paper our careful consideration and find it of potential interest. However, due to certain shortcomings we are concerned that sending the current manuscript out to review could lead to unnecessary delays and quite possibly an undesirable outcome of the review process.

We think the presentation of a new single-cell chromatin profiling method, incorporating FACS, would be of broad interest and certainly of importance to the journal.

Our team, however, thinks that there is one clear gap in your study as currently presented: a head-to-head comparison of sortChIC to the competing methods. As reviewers are - with near-certainty - likely to ask for this kind of methodological benchmarking, we think that this aspect could be productively addressed before peer review.

We would therefore like to invite you to revise your manuscript to address these concerns before we make a final determination on whether to send your manuscript for external review.

We shall hope to receive your revised version as soon as you are able to complete the suggested revisions. If something similar is published in the interim we will have to consider the impact it has on the novelty of a revised manuscript.

If you anticipate a delay of more than four weeks, please let us know. We will be happy to consider your revision so long as nothing similar has been accepted for publication at Nature Genetics or published elsewhere. Should your manuscript be substantially delayed without notifying us in advance and your article is eventually published, the received date may be that of the revised, not the original, version.

If you are not interested in submitting a suitably revised manuscript in the future please let me know immediately so we can close your file.

If you have any questions, please contact me - I am happy to discuss any issues further via email or a call.

1) Please ensure that you have completed the Reporting Summary required for review:
<https://www.nature.com/documents/nr-reporting-summary.pdf>

2) Please also complete the Editorial Policy Checklist that would be a requirement for eventual publication in a Nature journal:
<https://www.nature.com/documents/nr-editorial-policy-checklist.pdf>

Please use the link below when you are prepared to resubmit.
[redacted]

Thank you for your interest in Nature Genetics.

Sincerely,

Michael Fletcher, PhD
Associate Editor, Nature Genetics

ORCID: 0000-0003-1589-7087

Decision Letter, first revision:

11th Aug 2021

Dear Alexander,

Your Technical Report, "Hierarchical chromatin regulation during blood formation uncovered by single-cell sortChIC" has now been seen by 3 referees. You will see from their comments below that while they find your work of interest, some important points are raised. We are interested in the possibility

of publishing your study in Nature Genetics, but would like to consider your response to these concerns in the form of a revised manuscript before we make a final decision on publication.

Briefly, all three reviewers appreciate the technical aspects of sortChIC and the utility of your method. However, there remain comments on shortcomings regarding comparisons to other methods, or the biological novelty.

Reviewer #1 is the most positive; they don't have any major requests for further work. Their primary comment is regarding the two-mark analysis, asking for an expanded discussion of how this was done.

Reviewer #2 thinks that the biological novelty of your work could be improved. They are unsure regarding the meaningfulness of a differentiation trajectory defined by repressive chromatin. They make some important technical comments regarding the cell populations analysed, as well as suggestions for further experiments and in silico work that would improve this biological novelty.

Reviewer #3 has focused on the comparison of sortChIC to other methods; they say this must be greatly expanded.

We think that Reviewer #3's suggestion for a more thorough comparison to other methods is vital. We note that there are cases of overlapping comments that are especially important - for instance, the suggestion from Reviewers #2 and #3 to expand the computational analysis to integrate other (published) data types.

We also think that the biological novelty should be improved, for example via this integrative analysis, or as Reviewer #2 suggests, by expanding your analysis of the less-well characterised sub-populations of the HSPC pool and performing a "full pseudotime trajectory" analysis.

To guide the scope of the revisions, the editors discuss the referee reports in detail within the team, including with the chief editor, with a view to identifying key priorities that should be addressed in revision and sometimes overruling referee requests that are deemed beyond the scope of the current study. We hope that you will find the prioritized set of referee points to be useful when revising your study. Please do not hesitate to get in touch if you would like to discuss these issues further.

We therefore invite you to revise your manuscript taking into account all reviewer and editor comments. Please highlight all changes in the manuscript text file. At this stage we will need you to upload a copy of the manuscript in MS Word .docx or similar editable format.

*2) If you have not done so already please begin to revise your manuscript so that it conforms to our

Technical Report format instructions, available [here](http://www.nature.com/ng/authors/article_types/index.html). Refer also to any guidelines provided in this letter.

[redacted]

Sincerely,

Michael Fletcher, PhD
Associate Editor, Nature Genetics

ORCID: 0000-0003-1589-7087

Referee expertise: single-cell genomics methods, epigenetics. Referee #2 also has expertise in the haematopoietic system.

Reviewers' Comments:

Reviewer #1:

Remarks to the Author:

The manuscript "Hierarchical chromatin regulation during blood formation uncovered by single cell sortChIC" develops a high-yield method of profiling histone modifications in individual cells, and applies to a diverse population of cell types. Overall, this is a very interesting paper that is enjoyable to read, with important technical and bioinformatic advances. The paper demonstrates high-quality data from relatively large numbers of cells. The analysis of individual marks and how these differ between cell types is clear. I do have questions about the final section "Distinct cell types can share similar heterochromatin landscapes" that would be addressed by more discussion. The authors profile two marks simultaneously in a population of cells, and then use single-mark profiling data to assign chromatin features in the two-mark dataset. This is interesting as applied to cluster analysis, but I wonder how robust this is, and how different the probabilities for all possible models were. Are there strategies for confirming that the highest-probability model is correct? It also seems these probabilities could be driven by a very small number of genes, which would make feature selection critical. Also, how is this kind of deconvolution affected if cell types are very similar? Overall, I found this strategy very interesting and novel, and so would appreciate more discussion.

Minor points:

Line 87 states "...we process 1128 K562 cells in the G1 phase of the cell cycle to ensure a single genome copy per cell." K562 cells are triploid, so I am unsure what the authors mean to say here.

Line 214 states "...HSPCs often have intermediate levels of H3K4me1 and H3K4me3 (Fig 4a, left two panels), suggesting a generally more accessible chromatin state [in] HSPCs." This figure does not clearly show this result as HSPCs are not centrally located in H3K4me3 data. The point is more clear in the Supplementary Figure 4a, and perhaps this could be included in the main figures.

Line 221 states "...H3K27me3 and H3K9me3, mainly distinguish between HSPCs and differentiated cell types, thereby marking the progress along the differentiation trajectory (Fig. 4a, right two panels)." But this does not accurately represent the gain and loss of signals between HSPCs and cell types. Further, the differentiation trajectory here is not defined in this data, the meaning of "progress" is not clear. Perhaps I'm not seeing the utility of this figure, and the authors should more clearly explain what this figure shows.

Fig. 4b shows changes in marks between HSPCs and each cell type. This could be more informative to distinguish gains in all cell types, losses in all cell types, etc.

Figure 5 annotates the scale bars as "TF motif activity". It is not clear here what this means. The term comes from reference 39, which states "the activity A_m quantifies how much each occurrence of motif m contributes to the level of epigenetic mark M in sample s ." Without more explanation (perhaps at line 260), this term seems misleading. It seems this is just an association between a motif and a mark, and calling it 'activity' gives it a mechanistic veneer.

The single cell read displays in Fig 1b are clear. In all other figures these seem washed out. Is there a reason for this, or could the images be adjusted?

Reviewer #2:

Remarks to the Author:

In this study, Zeller et al. introduce sortCHIC to map histone modifications in single cells by utilizing antibodies recognizing select histone modifications followed by cleavage of adjacent DNA fragments by protein A-coupled micrococcal nuclease. The approach is compatible with cell enrichment using surface markers by FACS prior to cell lysis and library preparation, thus, enabling studies of histone modifications in rare or underrepresented cell populations.

In a proof-of-concept experiment in K562 cells, the authors demonstrate sortCHIC's sensitivity to detect the histone modifications H3K4me1, H3K4me3, H3K9me3 and H3K27me3 at the single cell level and show that pseudobulk analysis of sortCHIC-data obtains similar results as previously published (bulk) ChIP-seq experiments of the same histone modifications.

The authors then characterize these histone modifications in different murine bone marrow populations/fractions (unenriched, Lineage-negative and LSK). By analyzing the presence of activating (and depletion of repressing) histone modifications at cell type-specific genes, the authors annotated different cell populations and their frequencies among the previously (un)enriched fractions. The authors suggest that active histone modification profiles (H3K4me1, H3K4me3) are specific for cell types, while repressive histone modifications H3K9me3 and H3K27me3 may differ substantially from the initial pattern in hematopoietic stem and progenitor cells (HSPCs) but may be largely shared between different differentiated cell types, in particular within a single lineage as the myeloid lineage. Based on these observations, the authors conclude that repressive histone marks define differentiation trajectories, whereas activating histone modifications, determine specific cellular identities.

While similar approaches as also cited by the authors have been published, this is an interesting study as it sheds light on global patterns of histone modifications that contribute to lineage specification in the hematopoietic system and provides a technology to replicate such investigations in other biological systems at the single cell level. The manuscript is well written, and the technical work appears very well done. However, this reviewer is concerned about the study design and the validity of some of the biological concepts that are being carried forward.

Major comments:

1. This reviewer finds the concept of heterochromatin dynamics to "define a differentiation trajectory" or "establish a lineage" that is put forward throughout the manuscript, some examples below, difficult to follow and problematic.

Lines 221-223: "By contrast, repressive chromatin dynamics, marked by H3K27me3 and H3K9me3, mainly distinguish between HSPCs and differentiated cell types, thereby marking the progress along the differentiation trajectory."

Lines 229-230 "...suggesting that many changes in repressive chromatin during hematopoiesis occur independent of the specific cell fate."

Lines 331-334: "This joint analysis confirms that distinct cell types in related lineages can share their heterochromatin state (Fig. 6e, f), suggesting a hierarchical model where changes in heterochromatin establish lineages and changes in active chromatin define cell types within lineages."

In most cell types only a fraction of the genes are active and necessary to establish that cell type, and it doesn't seem unexpected that the large rest of the genome is being repressed by similar mechanisms and that the related changes in repressive chromatin are independent of cell fate. The point is also well taken that repressive chromatin in more closely related cell types, as within cells of the myeloid lineage, as opposed to erythroblasts or also neurons might be more similar. However, the presentation of "define" a trajectory or "establish" a lineage doesn't seem adequate in this reviewer's opinion. For example, if this as an "active" process were to be disrupted and also depending on the extent, how would differentiation be affected? In a hypothetical experiment, if one were able to switch or reprogram the repressive chromatin of a myeloid cell to that of an erythroblast, would that be sufficient to redefine the lineage? In particular, if the activating marks in the erythroblast stay the same? The role of activating marks to drive expression of genes seems a lot more crucial and intuitive to follow, but with respect to heterochromatin dynamics and their role to really define a lineage based on largely descriptive data, this reviewer would suggest using more neutral wording to communicate these global observations.

2. This reviewer also wonders to what extent the sampling strategy presents a potential confounder to many of the analysis and the conclusions that are being drawn from it. The unenriched cell fraction is vastly dominated by neutrophils, because of which many more differentiated cell types of interest, such as T cells and monocytes do not appear represented at all. The Lin negative enrichment would be further expected to deplete differentiated cells and it's a bit unclear to this reviewer what stage of B cells are indeed being captured in Fig. 2, as supposedly mature B cells should have been depleted. Ebf1 is also expressed in immature B cells (and CLPs) already. Also, the HSPC pool (LSK) is of course heterogenous and while the technology is impressive, in its current form it appears unable to further resolve cell types within. For example, for the analysis in Fig. S2d-f, statements as

"...suggesting that cell-type specific regions in HSPCs already have an intermediate level of active chromatin marks, which are then modulated up or down depending on the cell type into which the HSPCs differentiate",

are problematic, as these could present already committed progenitor cells, such as a MEPs, GMPs, or CLPs, that are also likely overrepresented compared to hematopoietic stem cells and multipotent progenitor cells, the chromatin states of which remain uncertain. In a sense, a form of pseudotime analysis along the full differentiation trajectory from the HSCs all the way through intermediate stages to several more differentiated cells (e.g. neutrophils, monocytes or erythroblasts) and a comparison across would present a much more holistic view of the global chromatin patterns, which the author are also aiming to describe and would be an incredibly valuable resource.

3. Independent of additional experiments and sampling of intermediate populations, are there computational strategies to improve the annotation of states of the existing data? Such as label transfer from RNA-seq datasets? Could a joint embedding using canonical correlation analysis, at least for H3k4me, H3K4me1, H3K27me3 marks facilitate more highly resolved annotation?

4. When comparing the performance of sortCHIC to other related methodologies, see Fig. S1g. How much could this be influenced by the cell type under investigation? Within the study, does the yield/quality vary when comparing K562 cells to the primary cells and here also within the different cell types?

5. The authors make an interesting observation relating to the association of distinct TF motifs with H3K27me3 and H3K9me3, see Fig. 5c,d. Can the authors show examples of genes/regions where these marks are indeed exclusive and how this may relate to the different roles and functions of each mark? Can the authors make any statements of how these marks may be of relevance to different progenitor cells within the HSPC pool? Are histone marks at these genes or their expression (based on scRNA-seq data) of the underlying TFs specific to a differentiation trajectory?

6. The method section of the sortChIC-seq experiments are a bit confusing and would benefit from more details. For example, the authors speak about sorting of plates at the end of the fixation and nuclei section and it's a bit unclear on how to proceed from one section to the next. Also, both mentioned wash buffers differ with respect to their composition, one using Tween, the other Saponin and if crucial they should be relabeled as buffer 1 and 2 or so as various washes with wash buffer follow throughout the protocols. It's also not clear which protease inhibitor cocktail and how much of it should be used.

7. For some examples of loci shown as for example in Fig. 4d and lines 236-239 in the main text, regarding the downregulation of H3K9me3 modifications in the Igh region, it would be beneficial to show an overview of the different histone marks. Notably, the Igh region appears relatively de-repressed in myeloid cells compared to HSPCs, nevertheless they likely differ with respect to their activating marks, in particular in the lymphoid cells, and it would be good to show for completion.

Minor comments:

3. Line 538: full stop is missing

4. Line 676: "...H3K27me..." should be "...H3K27me3..."

5. Figure 3b: Labels of clusters 3 and 4 are potentially switched unless this was intended and the match to the data is correct.

6. Line 213: "...where the erythroblasts shows..." should be "...where the erythroblasts show..."

7. On x-axis of 1st and 3rd panels of Fig. 4: "Ertyhs" should be "Eryths". Probably also best to keep a consistent label across the manuscript, such as erythroid.

8. Supplementary Figure 5b is designated as "5a".

Reviewer #3:

Remarks to the Author:

In Zeller & Yeung et al, led by van Oudenaarden, the authors present sortChIC as a method to probe histone modifications in single cells within the context of hematopoiesis. Overall, the manuscript is well written and the analyses performed are sound; however, the reviewers fail to properly document prior technologies that form the basis of their workflow modification (specifically the inclusion of sorting), nor do they compare their data to the most relevant other techniques. It would also benefit greatly from integration with other available datasets – right now they are analyzed in a vacuum and a number of exciting analysis that could yield important biological information are missing.

The technology itself is essentially CUT&RUN, with protein A – MNase bound to targeting antibodies followed by cleavage, which has been performed on single cells. Note – this is not the only ChIC-named technology – which is also essentially CUT&Run, though others are referenced (eg Ku et al 2019), but not described in the manuscript as other related technologies. How is sortChIC different? How does it compare? (see comments on supp fig 1g as well below).

There is also another paper: Ku et al 2021 Genome Research “Profiling single-cell histone modifications using indexing chromatin immunocleavage sequencing” that is not referenced at all; granted it is recent, but it should be compared.

While the technology is based on CUT&RUN without novel molecular techniques, just the addition of sorting, that should not take away from the impact. The optimization of the workflow is difficult and the authors ultimately present a high-quality protocol that appears to generate valuable data.

It is odd that there is no reference to CUT&RUN in the text or single-cell CUT&RUN (eg Hainer et al., 2019, Cell as one example). The technique is essentially the same, but a modified workflow to include sorting, so it is odd that there is no reference or direct comparison – just comparisons to CUT&Tag.

Supp Fig 1g. The comparison of FRiP across methods can be difficult. Was a unified set of peaks used? Or was it just the peaks called for each independent dataset? If the latter, then this metric is not appropriate, as the peak calling power can be directly associated with cell number & sequencing depth and not actual library quality. The same peak windows should be used to assess each method’s FRiP.

Supp Fig. 1g. Unique reads is influenced by sequencing saturation. If a library *could* achieve high read count but is not sequenced deeply (ie low saturation) then it will have a low unique read count. To control for this one can either: 1) subsample all libraries to be the same sequencing saturation as estimated by the % unique reads or 2) calculate the estimated total library size at complete saturation and compare those values. As it stands now there is no context in this plot. Also (as well as in the paper), it does not include single-cell CUT&RUN, which is a much more direct comparison.

There is a wealth of single-cell RNA-seq and single-cell ATAC-seq data on hematopoiesis yet those datasets are not leveraged here (though referenced, one as ref 10, though many more unreferenced datasets are available, much led by Greenleaf, Chang, and Buenrostro labs). As it stands an entire and important component of the analysis of these datasets are missing. There are powerful tools for dataset integration where these datasets could be leveraged to characterize the impact of histone modifications on transcriptional readout and broader chromatin state.

Author Rebuttal, first revision:**Response to Reviewer #1:**

Remarks to the Author:

The manuscript “Hierarchical chromatin regulation during blood formation uncovered by single cell sortChIC” develops a high-yield method of profiling histone modifications in individual cells, and applies

to a diverse population of cell types. Overall, this is a very interesting paper that is enjoyable to read, with important technical and bioinformatic advances. The paper demonstrates high-quality data from relatively large numbers of cells. The analysis of individual marks and how these differ between cell types is clear. I do have questions about the final section “Distinct cell types can share similar heterochromatin landscapes” that would be addressed by more discussion. The authors profile two marks simultaneously in a population of cells, and then use single-mark profiling data to assign chromatin features in the two-mark dataset. **This is interesting as applied to cluster analysis, but I wonder how robust this is, and how different the probabilities for all possible**

models were. Are there strategies for confirming that the highest-probability model is correct? It also seems these probabilities could be driven by a very small number of genes, which would make feature selection critical. Also, how is this kind of deconvolution affected if cell types are very similar? Overall, I found this strategy very interesting and novel, and so would appreciate more discussion.

We are happy to hear your supportive comments. To ask whether the double-incubation analysis is sensitive to feature selection, we redid the analysis using genome-wide bins rather than cell type-specific bins. We find the dual cluster inference gave very similar results whether we used genome-wide bins or select for cell type-specific bins (see figure below).

The probabilities of different models are often dominated by one unique model, suggesting that there is a statistical signal to assign a cell to a unique pair of clusters. In Supplemental Figure 7d we show

heatmaps of the log-likelihood, and when translating the log-likelihoods into probability assignments (by taking the normalized exponential, i.e. softmax) they all translate to nearly 100% assignment probability.

Generalizing these heatmaps to all cells by plotting the histogram of the highest probability, we find that most assignments are greater than 90% (Supplementary Fig. 7e):

The dual cluster inference relies on annotated clusters. If two clusters are very similar to each other (e.g., two Louvain clusters that actually belong to one cell type), one will see this reflected in the assignment matrix where one cell type would split into the two similar clusters.

Minor points:

Line 87 states "...we process 1128 K562 cells in the G1 phase of the cell cycle to ensure a single genome copy per cell." **K562 cells are triploid**, so I am unsure what the authors mean to say here.

Thank you for this comment. This description was aimed at primary cells, but indeed in the case of K562 cells, which are polyploid, this description is confusing. We changed this now to “to avoid a mixing of cell type and cell-cycle specific changes”.

Line 214 states “...HSPCs often have intermediate levels of H3K4me1 and H3K4me3 (Fig 4a, left two panels), suggesting a generally more accessible chromatin state [in] HSPCs.” **This figure does not clearly show this result as HSPCs are not centrally located in H3K4me3 data. The point is more clear in the Supplementary Figure 4a, and perhaps this could be included in the main figures.**

We agree with the reviewer that the original Fig. 4a did not show the intermediate levels of H3K4me1 and H3K4me3 in HSPCs very clearly. We followed the advice of Reviewer 1 and have now swapped Supplementary Figure 4a with Figure 4a.

In addition, we have now included new experiments that include FACS-validated populations of HSCs, different progenitor types and mature cell types. Pseudotime analysis of TSS and dynamic regions shows an intermediate chromatin state for H3K4me1 and H3K4me3 in HSCs (Figure 7e-f and Supplementary Figure 10 - 11).

We believe this pseudotime analysis across FACS-validated HSCs, progenitors, and differentiated cell types further validates that HSPCs have intermediate levels of H3K4me1 and H3K4me3 at cell type-specific loci.

Line 221 states “...H3K27me3 and H3K9me3, mainly distinguish between HSPCs and differentiated cell types, thereby marking the progress along the differentiation trajectory (Fig. 4a, right two panels).” **But this does not accurately represent the gain and loss of signals between HSPCs and cell types. Further, the differentiation trajectory here is not defined in this data, the meaning of “progress” is not clear. Perhaps I’m not seeing the utility of this figure, and the authors should more clearly explain what this figure shows. Fig. 4b shows changes in marks between HSPCs and each cell type. This could be more informative to distinguish gains in all cell types, losses in all cell types, etc.**

Similar to the comment before, we agree that Fig. 4a is not the best representation of the differentiation trajectories in H3K9me3 and H3K27me3. To systematically define differentiation trajectories, we now add new experiments that include FACS-validated populations of HSCs, different progenitor types and mature cell types. A pseudotime analysis of TSS and dynamic regions shows an intermediate chromatin state for H3K4me1 and H3K4me3 in HSCs, while for the repressive marks H3K27me3 and H3K9me3, the largest dynamics come from gain or loss of bins across all differentiated cell types. We have now included this trajectory analysis in Fig. 7 and Supplementary Fig. 10 - 11.

Figure 5 annotates the scale bars as “TF motif activity”. It is not clear here what this means. The term comes from reference 39, which states “the activity A_m quantifies how much each occurrence of motif m contributes to the level of epigenetic mark M in sample s .” Without more explanation (perhaps at line 260), this term seems misleading. It seems this is just an association between a motif and a mark, and calling it ‘activity’ gives it a mechanistic veneer.

Thank you for this comment. “TF motif activity” is a term often used in computational biology papers over the years (e.g. FANTOM Consortium *Nature Genetics* 2009, Carey et al Schacht et al *Bioinformatics* 2014, Garcia-Alonso et al *Genome Research* 2019), but we realized that we did not offer a sufficient explanation of this term and what we mean. We scanned the literature and we believe the computational idea and term “TF activity” stems from Bussemaker in Bussemaker et al 2001 *Nature Genetics* and subsequently the methods paper Roven and Bussemaker 2003 *Nucleic Acids Research*. Our intention to use the term “TF motif activity” was not to give a mechanistic veneer, but rather to help connect our method to other previous bulk methods that have tried to model the data in a similar manner.

The term has a more technical meaning, since it is used more often in computational biology papers. Specifically, here we use “TF motif activity” to model the chromatin dynamics across cells (our experimental measurement, which is a matrix of size g regions by c cells), as a function of the presence of *cis*-regulatory elements (which is the same in all cells but differs across genomic regions, represented by a matrix of size g regions by m motifs) and a hidden variable, which we call “TF motif activity” (TF activity of each differs in each cell, represented by a matrix of size m motifs by c cells). We infer the TF motif activity globally using ridge regression. We are therefore trying to predict statistically significant TF motifs that correlates with chromatin dynamics. A technical point here is that we are inferring significant TF motifs by inferring these hidden parameters in single cells, which we can then overlay the parameter estimates onto a UMAP, which greatly helps interpretability.

We have now added more explanation for the term TF motif activity in the results section.

The single cell read displays in Fig 1b are clear. In all other figures these seem washed out. Is there a reason for this, or could the images be adjusted?

Thank you for pointing this out. We replaced these plots with higher resolution versions.

Response to Reviewer #2:

Remarks to the Author:

In this study, Zeller et al. introduce sortCHIC to map histone modifications in single cells by utilizing antibodies recognizing select histone modifications followed by cleavage of adjacent DNA fragments by protein A-coupled micrococcal nuclease. The approach is compatible with cell enrichment using surface markers by FACS prior to cell lysis and library preparation, thus, enabling studies of histone modifications in rare or underrepresented cell populations.

In a proof-of-concept experiment in K562 cells, the authors demonstrate sortCHIC's sensitivity to detect the histone modifications H3K4me1, H3K4me3, H3K9me3 and H3K27me3 at the single cell level and show that pseudobulk analysis of sortCHIC-data obtains similar results as previously published (bulk) ChIP-seq experiments of the same histone modifications.

The authors then characterize these histone modifications in different murine bone marrow populations/fractions (unenriched, Lineage-negative and LSK). By analyzing the presence of activating (and depletion of repressing) histone modifications at cell type-specific genes, the authors annotated different cell populations and their frequencies among the previously (un)enriched fractions. The authors suggest that active histone modification profiles (H3K4me1, H3K4me3) are specific for cell types, while repressive histone modifications H3K9me3 and H3K27me3 may differ substantially from the initial pattern in hematopoietic stem and progenitor cells (HSPCs) but may be largely shared between different differentiated cell types, in particular within a single lineage as the myeloid lineage. Based on these observations, the authors conclude that repressive histone marks define differentiation trajectories, whereas activating histone modifications, determine specific cellular identities.

While similar approaches as also cited by the authors have been published, this is an interesting study as it sheds light on global patterns of histone modifications that contribute to lineage specification in the hematopoietic system and provides a technology to replicate such investigations in other biological systems at the single cell level. The manuscript is well written, and the technical work appears very well done. However, this reviewer is concerned about the study design and the validity of some of the biological concepts that are being carried forward.

Major comments:

1. This reviewer finds the concept of heterochromatin dynamics to **“define a differentiation trajectory”** or **“establish a lineage”** that is put forward throughout the manuscript, some examples below, difficult to follow and problematic.

Lines 221-223: “By contrast, repressive chromatin dynamics, marked by H3K27me3 and H3K9me3, mainly distinguish between HSPCs and differentiated cell types, thereby marking the progress along the differentiation trajectory.”

Lines 229-230 “...suggesting that many changes in repressive chromatin during hematopoiesis occur independent of the specific cell fate.”

Lines 331-334: “This joint analysis confirms that distinct cell types in related lineages can share their heterochromatin state (Fig. 6e, f), suggesting a hierarchical model where changes in heterochromatin establish lineages and changes in active chromatin define cell types within lineages.”

In most cell types only a fraction of the genes are active and necessary to establish that cell type, and it doesn't seem unexpected that the large rest of the genome is being repressed by similar mechanisms and that the related changes in repressive chromatin are independent of cell fate. The point is also well taken that repressive chromatin in more closely related cell types, as within cells of the myeloid lineage, as opposed to erythroblasts or also neurons might be more similar. **However, the presentation of “define” a trajectory or “establish” a lineage doesn't seem adequate in this reviewer's opinion. For example, if this as an “active” process were to be disrupted and also depending on the extent, how would differentiation be affected? In a hypothetical experiment, if one were able to switch or reprogram the repressive chromatin of a myeloid cell to that of an erythroblast, would that be sufficient to redefine the lineage? In particular, if the activating**

marks in the erythroblast stay the same? The role of activating marks to drive expression of genes seems a lot more crucial and intuitive to follow, but with respect to heterochromatin dynamics and their role to really define a lineage based on largely descriptive data, this reviewer would suggest using more neutral wording to communicate these global observations.

We thank the reviewer for these comments. Indeed, this manuscript does not contain any functional experiments and our choice of words could be misinterpreted. We now changed the phrasing in the mentioned paragraphs.

2. This reviewer also wonders to what extent the sampling strategy presents a potential confounder to many of the analysis and the conclusions that are being drawn from it. **The unenriched cell fraction is vastly dominated by neutrophils, because of which many more differentiated cell types of interest, such as T cells and monocytes do not appear represented at all. The Lin negative enrichment would be further expected to deplete differentiated cells and it's a bit unclear to this reviewer what stage of B cells are indeed being captured in Fig. 2, as supposedly mature B cells should have been depleted.** Ebf1 is also expressed in immature B cells (and CLPs) already. Also, the HSPC pool (LSK) is of course heterogenous and while the technology is impressive, in its current form it appears unable to further resolve cell types within. For example, for the analysis in Fig. S2d-f, statements as

“...suggesting that cell-type specific regions in HSPCs already have an intermediate level of active chromatin marks, which are then modulated up or down depending on the cell type into which the HSPCs differentiate”,

are problematic, as these could present already committed progenitor cells, such as a MEPs, GMPs, or CLPs, that are also likely overrepresented compared to hematopoietic stem cells and multipotent progenitor cells, the chromatin states of which remain uncertain. In a sense, a form of pseudotime analysis along the full differentiation trajectory from the HSCs all the way through intermediate stages to several more differentiated cells (e.g. neutrophils, monocytes or erythroblasts) and a comparison across would present a much more holistic view of the global chromatin patterns, which the author are also aiming to describe and would be an incredibly valuable resource.

Thank you for this comment. We agree that the unenriched bone marrow is dominated by a few cell types making a broad description of cell types difficult. To dissect committed progenitors, such as MEPs, and perform a full pseudotime analysis along the full differentiation trajectory, we generated an expanded sortChIC dataset. This new dataset includes cell types along HSCs, intermediate stages, to differentiated cell types. Importantly, this expanded dataset includes surface marker stainings to sort specific progenitor cell types (i.e., HSCs, LTs, STs, MPPs, CMPs, MEPs) as well as record surface markers (i.e., Sca1, cKit, and Lineage) that can be used to determine the maturation stage of cells.

With this expanded sortChIC dataset, we have increased the number of cells by about 70% from 22587 to 39857 cells (across H3K4me1, H3K4me3, H3K27me3, and H3K9me3 modifications). These labeled cells also enable a supervised analysis to refine our cell typing analysis to distinguish subtypes with HSPCs as well as validate differentiated cell types (specifically B cells, NK cells, erythroblasts, neutrophils, monocytes, pDCs, and DCs) across histone modifications. Finally, from this new dataset, we have now performed a pseudotime analysis from HSCs to intermediate stages and mature cell types. For example, we find cell type-specific marker genes increasing for active marks (H3K4me1 and H3K4me3) while decreasing for H3K27me3. Furthermore, we find hematopoietic stem cell markers decrease in H3K4me1 and H3K27me3 while decreasing for H3K27me3. Finally, we use the Sca1, cKit, and Lineage markers to define a common differentiation time across the four histone modification datasets and find that lineage specification at the global chromatin level occurs after around the peak of cKit levels, relative to Sca1 and Lineage. We have incorporated this pseudotime analysis into the manuscript in Figure 7 and Supplemental Figure 10-11.

3. Independent of additional experiments and sampling of intermediate populations, are there computational strategies to improve the annotation of states of the existing data? **Such as label transfer from RNA-seq datasets? Could a joint embedding using canonical correlation analysis, at least for H3k4me, H3K4me1, H3K27me3 marks facilitate more highly resolved annotation?**

Thank you for this suggestion. We note that the goal of joint embedding methods such as canonical correlation analysis is to have a single UMAP (or latent space) incorporate different datasets. While this assumption has been successful for integrating scRNA-seq and scATAC-seq data, we note that one of our major conclusions is that the repressive marks, H3K27me3 and H3K9me3, have qualitatively distinct cell-cell relationships compared with active marks. It is still an open question of how to learn joint embeddings on datasets that have nontrivial (all positively correlated or all negatively correlated) relationships between cell types.

We tried to improve cell type annotations using canonical correlation analysis between H3K4me1 and scRNA-seq data with recently published scRNA-seq data of the bone marrow, Baccin et al 2020 *Nature Cell Biology* that used a comparable enrichment strategy. This integrated analysis further corroborated our cell typing analysis, such as erythroblasts, neutrophils, B cells, and NK cells. Interestingly, the scRNA-seq data revealed that the T cells are transcriptionally similar to NK cells and occupy overlapping regions in the joint UMAP. This intermingling suggests that the T cells in our dataset would likely mix in with our NK cells, although our unsupervised analysis was not able to separate the two cell types.

Concerning sampling of intermediate populations, we were not able to find subclusters within our HSPCs. To see whether our original dataset had significant intermediate populations, we generated an expanded bone marrow dataset that explicitly includes intermediate populations such as CMPs and MEPs, and then compared this new dataset with our original dataset. We find that the expanded dataset includes more CMPs and MEPs compared to the original data, suggesting that our initial identification of the HSPC cluster did not include many intermediate populations.

4. When comparing the performance of sortCHIC to other related methodologies, see Fig. S1g. How much could this be influenced by the cell type under investigation? Within the study, does the yield/quality vary when comparing K562 cells to the primary cells and here also within the different cell types?

The comparison of sortChIC versus other methodologies is unfortunately limited by the cell types used in the experiments, histone marks targeted as well as antibodies used. The comparisons can be influenced by different cell lines and different primary tissues. We now use more comparable metrics across studies by focusing on H3K27me3 (a mark that is common to most studies), by separating between cell lines and primary cells, and by using metrics such as Gini coefficient which does not rely on peak calling (Supplementary Fig. 2).

5. The authors make an interesting observation relating to the association of distinct TF motifs with H3K27me3 and H3K9me3, see Fig. 5c,d.

Point 5 consists of three different comments, so we have split our responses to address each comment separately.

Can the authors show examples of genes/regions where these marks are indeed exclusive and how this may relate to the different roles and functions of each mark?

The mutual exclusiveness of H3K27me3 and H3K9me3 is on a large A-B compartment level (scale of 1-10 Mb). The biggest features of these domains are that H3K27me3 tends to be GC-rich while H3K9me3 tends to be GC-poor (shown already previously in Supplemental Figure 5c,d), suggesting that the two repressive marks are located in different regions of the genome. To show this visually on the genome browser as an example, we show below that these exclusive regions in a 20 Mb window across cell types.

Can the authors make any statements of how these marks may be of relevance to different progenitor cells within the HSPC pool?

Subclustering the HSPC cells did not reveal biologically relevant clusters, suggesting that our HSPC cluster did not contain significant intermediate populations. We validated this by performing a new sortChIC dataset where we FACS-sorted HSCs, LTs, STs, MPPs, and committed progenitors (CMPs and

MEPs), and find that our HSPC pool did not intermingle with committed progenitors such as MEPs and CMPs. We now show the changes of both marks during the progenitor cell stage in Fig. 7.

Are histone marks at these genes or their expression (based on scRNA-seq data) of the underlying TFs specific to a differentiation trajectory?

Our H3K27me3 and H3K9me3 motif analysis predicted *Yy1* and *Plzf/Zbtb16* mRNA levels to have some cell type-specific expression. We plotted the gene expression of these two genes in the scRNA-seq of Baccin et al. We find only low expression of these two genes in the scRNA-seq data. This was unexpected since *Yy1* and *Plzf* have both been studied in mouse hematopoietic stem cells (e.g., Lu et al 2018 *Cell Reports*, Vincent-Fabert et al 2016 *Blood*). The low/no expression in the scRNA-seq dataset could be due to the low sensitivity of scRNA-seq or the HSPCs in Baccin et al. did not contain *Yy1* or *Plzf* expressing stem cells.

We then looked at H3K4me1 levels to see whether regions coding for *Yy1* and *Zbtb16* had active marks across cell types (see genome browser view below). We found that *Yy1* and *Zbtb16* had active marks in all cell types, at varying levels. For example, the erythroblast and B cells lose their H3K4me1 marks around *Zbtb16*. We note that the relationship between mRNA or transcription of a gene coding for a TF is not necessarily directly related with the TF's effect on up/down-regulating local epigenetic marks, so a direct relationship is not always observed.

6. The method section of the sortChIC-seq experiments are a bit confusing and would benefit from more details. For example, the authors speak about sorting of plates at the end of the fixation and nuclei section and it's a bit unclear on how to proceed from one section to the next. Also, both mentioned wash buffers differ with respect to their composition, one using Tween, the other Saponin and if crucial they should be labeled as buffer 1 and 2 or so as various washes with wash buffer follow throughout the protocols. It's also not clear which protease inhibitor cocktail and how much of it should be used.

We apologize for the method details not being clear enough written down. We rewrote that part and numbered consecutive steps for more clarity. We hope it has become more understandable like this.

7. For some examples of loci shown as for example in Fig. 4d and lines 236-239 in the main text, regarding the downregulation of H3K9me3 modifications in the Igh region, it would be beneficial to show an overview of the different histone marks. Notably, the Igh region appears relatively de-repressed in myeloid cells compared to HSPCs, nevertheless they likely differ with respect to their activating marks, in particular in the lymphoid cells, and it would be good to show for completion.

We have added H3K4me3 data to Fig. 4d. For the H3K27me3 regulated Hox cluster, we can indeed see the expected upregulation of the locus in HSCs and pDCs. In the case of the Igh locus, despite its loss of H3K9me3 the region does not show an upregulation in myeloid or lymphoid cells. This suggests that the B cells recovered in our assay are mainly at the pro-B cell stage, before Igh expression.

Minor comments:

3. Line 538: full stop is missing

4. Line 676: "...H3K27me..." should be "...H3K27me3..."

5. Figure 3b: Labels of clusters 3 and 4 are potentially switched unless this was intended and the match to the data is correct.

6. Line 213: "...where the erythroblasts shows..." should be "...where the erythroblasts show..."

7. On x-axis of 1st and 3rd panels of Fig. 4: "Ertyhs" should be "Eryths". Probably also best to keep a consistent label across the manuscript, such as erythroid.

8. Supplementary Figure 5b is designated as "5a".

We thank the reviewer for his detailed comments. We corrected all but the labeling in Figure 3b which was intentional.

Response to Reviewer #3:

Remarks to the Author:

In Zeller & Yeung et al, led by van Oudenaarden, the authors present sortChIC as a method to probe histone modifications in single cells within the context of hematopoiesis. Overall, the manuscript is well written and the analyses performed are sound; however, the reviewers fail to properly document prior technologies that form the basis of their workflow modification (specifically the inclusion of sorting), nor do they compare their data to the most relevant other techniques. It would also benefit greatly from integration with other available datasets – right now they are analyzed in a vacuum and a number of exciting analysis that could yield important biological information are missing.

The technology itself is essentially CUT&RUN, with protein A – MNase bound to targeting antibodies followed by cleavage, which has been performed on single cells. Note – this is not the only ChIC-named technology – which is also essentially CUT&Run, though others are referenced (eg Ku et al 2019), but not described in the manuscript as other related technologies. How is sortChIC different? How does it compare? (see comments on supp fig 1g as well below).

There is also another paper: Ku et al 2021 Genome Research “Profiling single-cell histone modifications using indexing chromatin immunocleavage sequencing” that is not referenced at all; granted it is recent, but it should be compared.

Thank you for highlighting the recent paper. Although we did cite this paper in the introduction (ref 27), we did not yet have it in the method comparison. We now added the data to the method comparison (Supplementary Figure 2).

While the technology is based on CUT&RUN without novel molecular techniques, just the addition of sorting, that should not take away from the impact. The optimization of the workflow is difficult and the authors ultimately present a high-quality protocol that appears to generate valuable data.

It is odd that there is no reference to CUT&RUN in the test or single-cell CUT&RUN (eg Hainer et al., 2019, Cell as one example). The technique is essentially the same, but a modified workflow to include sorting, **so it is odd that there is no reference or direct comparison** – just comparisons to CUT&Tag.

Although we did cite both papers in the introduction (ref. 26 and ref. 28) we did indeed not make a clear separation between CUT&RUN and other proteinA-MN dependent methods. We added now CUT&RUN as well as CUT&TAG specifically to the text. However, in our comparisons we focused on single-cell profiling of histone modifications. The single-cell profiling performed in Hainer et al 2019 was on pluripotency transcription factors. We therefore could not include this single-cell CUT&RUN dataset into our comparison in Supplemental Figure 2.

Supp Fig 1g. The comparison of FRiP across methods can be difficult. Was a unified set of peaks used? Or was it just the peaks called for each independent dataset? If the latter, then this metric is not appropriate, as the peak calling power can be directly associated with cell number & sequencing depth and not actual library quality. **The same peak windows should be used to assess each method's FRiP.**

We could not use a unified peak set in our comparison as different labs used different cell lines and cell types, as well as possibly different antibodies to target histone modifications. We therefore had to call peaks for each independent dataset. We agree with the reviewers' concern that peak calling accuracy can vary between datasets. To avoid confounding differences between datasets we now changed the FRiP score for two other metrics that do not rely on peak calling. First, we quantify the fraction of reads falling into the top 25% genomic bins. Second, we use Gini coefficient to quantify the unevenness of the signal across the genome (higher Gini means more signal is located at fewer bins).

Supp Fig. 1g. Unique reads is influenced by sequencing saturation. If a library *could* achieve high read count but is not sequenced deeply (ie low saturation) then it will have a low unique read count. To control for this one can either: 1) subsample all libraries to be the same sequencing saturation as estimated by the % unique reads or 2) calculate the estimated total library size at complete saturation and compare those values. As it stands now there is no context in this plot. Also (as well as in the paper), **it does not include single-cell CUT&RUN, which is a much more direct comparison.**

As stated above we can unfortunately not compare our data with single cell CUT&RUN from Hainer et al 2019 as there is no overlap in terms of antibody target. We have now included scChIC-seq (Ku et al.

2019) and iscChIC-seq (Lim Ku et al. 2021) to the detailed method comparison that can also both be considered single cell CUT&RUN approaches.

We agree that sequencing depth can be a confounder when assessing the sensitivity of an approach. We have now added a graph visualizing the ratio of total mapped reads vs the fraction of unique reads to compare PCR duplication rates. We could observe that besides Ku et al. 2021, which has a very high fraction of unique reads, all other studies were performed in a comparable range of sequencing depths.

We have now updated Supplemental Figure 2.

There is a wealth of single-cell RNA-seq and single-cell ATAC-seq data on hematopoiesis yet those datasets are not leveraged here (though referenced, one as ref 10, though many more unreferenced datasets are available, much led by Greenleaf, Chang, and Buenrostro labs). As it stands an entire and important component of the analysis of these datasets are missing.

There are powerful tools for dataset integration where these datasets could be leveraged to characterize the impact of histone modifications on transcriptional readout and broader chromatin state.

We note that many methods to integrate single-cell datasets, such as canonical correlation analysis, rely on an implicit assumption that there should be one common UMAP or latent space on which the different datasets should share. However, one of our main results is that the repressive marks H3K27me3 and H3K9me3 have qualitatively distinct dynamics compared to the active marks, and therefore should be represented on a separate UMAP.

We focused, therefore, on integrating publicly available data with the active histone mark, H3K4me1. We have now added two integrative analyses, combining mouse scRNA-seq with sortChIC and scATAC-seq with H3K4me1 sortChIC. We note that many of the single-cell hematopoiesis datasets led by Greenleaf, Chang, and Buenrostro labs were done in human hematopoiesis, and therefore we did not integrate these datasets with mouse data.

We found excellent integration of scRNA-seq with our H3K4me1 sortChIC dataset. We used this analysis to identify previously unidentified cell types, such as T cells, and validate many of our cell type analysis.

We also integrated our data with scATAC-seq mouse bone marrow dataset generated from the Shendure group (Cusanovich et al 2018 *Cell*). We integrated our H3K4me1 sortChIC dataset and found they had more cell types missing compared to ours (e.g., surprisingly, neutrophils were missing in their dataset), and there were discrepancies of cell types called by Cusanovich et al versus our study.

We have therefore focused our efforts to reconcile these differences and have generated a FACS-validated and comprehensive sortChIC dataset in Figure 7. Of note, this expanded dataset allowed us to perform an integrative analysis along differentiation potential (defined by relative levels of Sca1, cKit, Lineage levels), which overcomes the assumptions of a correlation structure between datasets which is implicit in many computational integrative methods. We have now shown this in Figure 7 and Supplemental Fig. 8-12.

Decision Letter, second revision:

18th Jul 2022

Dear Alexander,

Your Technical Report, "Hierarchical chromatin regulation during blood formation uncovered by single-cell sortChIC" has now been seen by the original 3 referees. You will see from their comments below that while they continue to find your work of interest, some important points still remain to be addressed.

We are interested in the possibility of publishing your study in Nature Genetics, but would like to consider your response to these concerns in the form of a revised manuscript before we make a final decision on publication.

Briefly, the reviewers all seem broadly satisfied with your responses, albeit to varying degrees. There are still a few comments remaining that we think need to be addressed before the final decision: specifically, Reviewer #2's comments on the (lack of) observed heterogeneity. We note that Reviewer #3 comes across as disappointed in some aspects of your responses, describing it as the "minimum"; we think it would be the best course of action to try and improve the manuscript further, but we do not think substantial further work should be required given the late stage of review and the overall level of referee support.

To guide the scope of the revisions, the editors discuss the referee reports in detail within the team, including with the chief editor, with a view to identifying key priorities that should be addressed in revision and sometimes overruling referee requests that are deemed beyond the scope of the current study. We hope that you will find the prioritized set of referee points to be useful when revising your study. Please do not hesitate to get in touch if you would like to discuss these issues further.

We therefore invite you to revise your manuscript taking into account all reviewer and editor comments. Please highlight all changes in the manuscript text file. At this stage we will need you to upload a copy of the manuscript in MS Word .docx or similar editable format.

*2) If you have not done so already please begin to revise your manuscript so that it conforms to our Technical Report format instructions, available [here](http://www.nature.com/ng/authors/article_types/index.html). Refer also to any guidelines provided in this letter.

[redacted]

We hope to receive your revised manuscript within four to eight weeks. If you cannot send it within this time, please let us know.

Nature Genetics is committed to improving transparency in authorship. As part of our efforts in this direction, we are now requesting that all authors identified as ‘corresponding author’ on published papers create and link their Open Researcher and Contributor Identifier (ORCID) with their account on the Manuscript Tracking System (MTS), prior to acceptance. ORCID helps the scientific community achieve unambiguous attribution of all scholarly contributions. You can create and link your ORCID from the home page of the MTS by clicking on ‘Modify my Springer Nature account’. For more information please visit www.springernature.com/orcid.

Sincerely,

Michael Fletcher, PhD
Senior Editor, Nature Genetics

ORCID: 0000-0003-1589-7087

Reviewers' Comments:

Reviewer #1:

Remarks to the Author:

The revised manuscript has added data which strengthens the conclusions and has addressed my questions on interpretations.

Reviewer #2:

Remarks to the Author:

The revised manuscript by Zeller and colleagues is substantially improved and most of my comments have been addressed in a satisfying manner. Overall, this is a high-quality manuscript, which should be considered for publication. However, this reviewer still has the following comments and would appreciate their consideration:

1. The new data is much appreciated and substantially adds to the manuscript. Nevertheless, its presentation and integration into the manuscript could be much more harmonized. Data/results related to Figures 2-4 would in fact benefit from the more highly resolved HSPC populations, which are now part of Fig. 7. Earlier integration would thereby also overall streamline the presentation of the manuscript.
2. The intermediate chromatin state of HSCs and progenitors are well taken. Nevertheless, based on published single cell omic datasets in hematopoiesis, this reviewer would have expected to resolve additional heterogeneity at the HSC/MPP/CMP stages. Potentially, this has not been more directly investigated by the authors, but if this cannot be resolved, the authors should also expand/discuss the limitations of the current implementation of their technique in more detail. This reviewer can only speculate, but potentially finer cell type/state annotations are not suitable with the marks that are being profiled / or with the current sensitivity of the technique, providing an area for future improvement, for example via integration of additional omic layers.

3. Middle Panels in Supplementary Fig. 10 c,d appear to be swapped?

4. The following lines would benefit from more neutral wording as touched upon in comment 1 of the original submission and as the authors responded was revised. However, the text was not updated it appears: Original submission Lines 331-334, Revised edition Lines 355-358: “This joint analysis confirms that distinct cell types in related lineages can share their heterochromatin state (Fig. 6e, f), suggesting a hierarchical model where changes in heterochromatin establish lineages and changes in active chromatin define cell types within lineages.”

Line 374: Capitalize “we”

Reviewer #3:

Remarks to the Author:

The authors address the majority of my comments. While I still believe a more robust comparison with other methods should be performed, the authors satisfy the minimum. While the authors are correct that many datasets are not perfect for comparison, there is never a perfect comparison and the best approximations must be used – this is what every new technique does to show its power.

On a positive note – the fraction of reads in top 25% genomic bins is an interesting metric. It would be greatly beneficial if the authors included it for additional percentages – e.g. top 10% and top 5%; top 25% appears a bit squished and a lower percent would spread out the data a bit more.

Regarding dataset integration: Yes it is true that different modalities show distinct UMAPS – that is the entire point of integration, so that the commonalities of the two modalities may emphasize new feature sets and enable better resolution than either mark on its own.

Author Rebuttal, second revision:

Reviewer #1:

Remarks to the Author:

The revised manuscript has added data which strengthens the conclusions and has addressed my questions on interpretations.

We are very glad that Reviewer 1 appreciates our revised manuscript describing our new experiments and recommends our manuscript for publication.

Reviewer #2:

Remarks to the Author:

The revised manuscript by Zeller and colleagues is substantially improved and most of my comments have been addressed in a satisfying manner. Overall, this is a high-quality manuscript, which should be considered for publication.

We thank Reviewer 2 for this positive feedback.

However, this reviewer still has the following comments and would appreciate their consideration:

1. The new data is much appreciated and substantially adds to the manuscript. Nevertheless, its presentation and integration into the manuscript could be much more harmonized. Data/results related to Figures 2-4 would in fact benefit from the more highly resolved HSPC populations, which are now part of Fig. 7. Earlier integration would thereby also overall streamline the presentation of the manuscript.

Figures 2-6 mostly focus on cell type differences relative to HSPCs. Our new data added the intermediate populations such as CMPs, MEPs and MPPs. To illustrate this, we show the UMAPs of the H3K4me3 data with Figure 2 data only, new data only, and merged data (i.e. Figure 7 data):

Since the new data is not providing new differentiated states or new stem cell states at the chromatin level, we do not believe that re-analyzing the data and remaking Figures 2-6 would substantially improve the manuscript at this late stage of review. This would require substantial re-designing of certain figures.

For example, Figure 3 and 6 currently detail how we determined cluster identities from H3K9me3 data, which we found to be challenging because no distinct marker genes are known. We believe that this analysis would be useful for the community who may be interested in annotating clusters from H3K9me3 data. Our current presentation of the manuscript allows us to discuss how we analyzed the H3K9me3 clusters without ground truth data (using antibodies). Our new data validates and further resolves these clusters to provide a high-confidence ground truth bone marrow dataset. Therefore, incorporating the final ground truth dataset into Figure 2 would likely make Figure 3 and 6 redundant, which we believe would not improve the paper.

2. The intermediate chromatin state of HSCs and progenitors are well taken. Nevertheless, based on published single cell omic datasets in hematopoiesis, this reviewer would have expected to resolve additional heterogeneity at the HSC/MPP/CMP stages. Potentially, this has not been more directly investigated by the authors, but if this cannot be resolved, the authors should also expand/discuss the limitations of the current implementation of their technique in more detail. This reviewer can only speculate, but potentially finer cell type/state annotations are not suitable with the marks that are being profiled / or with the current sensitivity of the technique, providing an area for future improvement, for example via integration of additional omic layers.

We thank the reviewer for this comment. We have explored the heterogeneity between the HSCs and different progenitor stages. Unfortunately, we were not able to identify clear differences at the chromatin level between these cell states. As suggested by the reviewer we now discuss the limitations of the current technology in the Discussion section of the manuscript.

3. Middle Panels in Supplementary Fig. 10 c,d appear to be swapped?

We thank the reviewer for pointing this out. The H3K4me1 panels were indeed swapped and are now corrected.

4. The following lines would benefit from more neutral wording as touched upon in comment 1 of the original submission and as the authors responded was revised. However, the text was not updated it appears: Original submission Lines 331-334, Revised edition Lines 355-358: "This joint analysis confirms that distinct cell types in related lineages can share their heterochromatin state (Fig. 6e, f), suggesting a hierarchical model where changes in heterochromatin establish lineages and changes in active chromatin define cell types within lineages."

We have now changed this particular sentence to: "This joint analysis confirms that distinct cell types in related lineages can share their heterochromatin state (Fig. 6e, f), suggesting a hierarchical model where

changes in heterochromatin might restrict lineages and changes in active chromatin define cell types within lineages.”

Line 374: Capitalize “we”

We have corrected this.

Reviewer #3:

Remarks to the Author:

The authors address the majority of my comments. While I still believe a more robust comparison with other methods should be performed, the authors satisfy the minimum. While the authors are correct that many datasets are not perfect for comparison, there is never a perfect comparison and the best approximations must be used – this is what every new technique does to show its power.

On a positive note – the fraction of reads in top 25% genomic bins is an interesting metric. It would be greatly beneficial if the authors included it for additional percentages – e.g. top 10% and top 5%; top 25% appears a bit squished and a lower percent would spread out the data a bit more.

We thank the Reviewer for these additional suggestions. We have now extended this part of the method comparison to include a full linear scale, ranging from 2.5-100% of the top covered bins. This indeed made this analysis even more informative. The new analysis is integrated in Supplementary Fig 2c-d.

Decision Letter, third revision:

16th Aug 2022

Dear Alexander,

Thank you for submitting your revised manuscript "Hierarchical chromatin regulation during blood formation uncovered by single-cell sortChIC" (NG-TR57651R2). It has now been seen by the original referees and their comments are below. The reviewers find that the paper has improved in revision, and therefore we'll be happy in principle to publish it in Nature Genetics, pending minor revisions to satisfy the referees' final requests and to comply with our editorial and formatting guidelines.

****Please note****: as the current version of your manuscript is in a PDF format, please email us a copy of the file in an editable format (Microsoft Word or LaTeX)-- ****we can not proceed with PDFs at this stage.****

Thank you again for your interest in Nature Genetics. Please do not hesitate to contact me if you have any questions.

Sincerely,

Michael Fletcher, PhD
Senior Editor, Nature Genetics

ORCID: 0000-0003-1589-7087

Final Decision Letter:

1st Nov 2022

Dear Alexander,

I am delighted to say that your manuscript "Single-cell sortChIC identifies hierarchical chromatin dynamics during hematopoiesis" has been accepted for publication in an upcoming issue of Nature Genetics.

Your paper will be published online after we receive your corrections and will appear in print in the next available issue. You can find out your date of online publication by contacting the Nature Press Office (press@nature.com) after sending your e-proof corrections. Now is the time to inform your Public Relations or Press Office about your paper, as they might be interested in promoting its publication. This will allow them time to prepare an accurate and satisfactory press release. Include your manuscript tracking number (NG-TR57651R3) and the name of the journal, which they will need

when they contact our Press Office.

Please note that *Nature Genetics* is a Transformative Journal (TJ). Authors may publish their research with us through the traditional subscription access route or make their paper immediately open access through payment of an article-processing charge (APC). Authors will not be required to make a final decision about access to their article until it has been accepted. [Find out more about Transformative Journals](https://www.springernature.com/gp/open-research/transformative-journals)

Authors may need to take specific actions to achieve [compliance](https://www.springernature.com/gp/open-research/funding/policy-compliance-faqs) with funder and institutional open access mandates. If your research is supported by a funder that requires immediate open access (e.g. according to [Plan S principles](https://www.springernature.com/gp/open-research/plan-s-compliance)) then you should select the gold OA route, and we will direct you to the compliant route where possible. For authors selecting the subscription publication route, the journal's standard licensing terms will need to be accepted, including [self-archiving-and-license-to-publish](https://www.nature.com/nature-portfolio/editorial-policies/self-archiving-and-license-to-publish). Those licensing terms will supersede any other terms that the author or any third party may assert apply to any version of the manuscript.

Please note that Nature Portfolio offers an immediate open access option only for papers that were first submitted after 1 January, 2021.

If you have not already done so, we invite you to upload the step-by-step protocols used in this manuscript to the Protocols Exchange, part of our on-line web resource, natureprotocols.com. If you complete the upload by the time you receive your manuscript proofs, we can insert links in your article that lead directly to the protocol details. Your protocol will be made freely available upon publication of your paper. By participating in natureprotocols.com, you are enabling researchers to more readily reproduce or adapt the methodology you use. [Natureprotocols.com](http://natureprotocols.com) is fully searchable, providing your protocols and paper with increased utility and visibility. Please submit your protocol to <https://protocolexchange.researchsquare.com/>. After entering your [nature.com](http://www.nature.com) username and password you will need to enter your manuscript number (NG-TR57651R3). Further information can be found at <https://www.nature.com/nature-portfolio/editorial-policies/reporting-standards#protocols>

Sincerely,

Michael Fletcher, PhD
Senior Editor, Nature Genetics

ORCID: 0000-0003-1589-7087

Click here if you would like to recommend Nature Genetics to your librarian
<http://www.nature.com/subscriptions/recommend.html#forms>

** Visit the Springer Nature Editorial and Publishing website at http://editorial-jobs.springernature.com?utm_source=ejP_NGen_email&utm_medium=ejP_NGen_email&utm_campaign=ejP_NGen for more information about our career opportunities. If you have any questions please click [here](mailto:editorial.publishing.jobs@springernature.com). **